# MVG-CRPS: A Robust Loss Function for Multivariate Probabilistic Forecasting

## Abstract

Multivariate probabilistic forecasting typically leverages neural network-based distributional regression, often employing Gaussian assumptions to simplify computation. While the standard negative log-likelihood provides analytical convenience, its sensitivity to outliers can severely degrade forecasting accuracy. Conversely, robust alternatives like the energy score, although less sensitive to extreme values, rely heavily on computationally expensive sampling approximations, limiting scalability in neural network training. To bridge this gap, we introduce the MVG-CRPS, a novel, strictly proper scoring rule for multivariate Gaussian distributions that maintains robustness to outliers while providing a closed-form expression, enabling efficient training and evaluation. Our approach leverages a whitening transformation, decorrelating multivariate outputs and reducing the multivariate scoring task to tractable univariate CRPS computations. Experiments on real-world datasets for both multivariate autoregressive and univariate sequence-to-sequence (Seq2Seq) forecasting tasks demonstrate that MVG-CRPS enhances robustness and predictive performance.

## 1 Introduction

Probabilistic forecasting is critical in applications ranging from financial risk management (Groen et al., 2013), to weather forecasting (Palmer, 2012) and healthcare analytics (Jones & Spiegelhalter, 2012), where accurate quantification of predictive uncertainty directly informs decision-making. Multivariate probabilistic forecasting models extend beyond point estimates, producing joint probability distributions across multiple correlated continuous variables. Neural network-based methods have become a dominant paradigm due to their flexibility and expressiveness (Salinas et al., 2019; Benidis et al., 2022; Rasul et al., 2021b). Typically, these methods rely on parametric assumptions such as multivariate Gaussian distributions, allowing closed-form loss computations (e.g., log-likelihood) and efficient backpropagation.

Despite widespread adoption, standard metrics for model inference such as the negative log-likelihood (log-score) present substantial challenges. Most notably, under the Gaussian family, the log-score heavily penalizes unlikely events and outliers due to its exponential sensitivity in the tails of distributions, making it excessively sensitive to anomalies and model misspecification (Gebetsberger et al., 2018; Bjerregård et al., 2021). As a result, neural network models trained using the log-score can generate overly conservative or inaccurate predictive distributions when exposed to real-world data characterized by occasional extreme events.

To address the limitations of the log-score, the energy score (ES, Gneiting & Raftery, 2007) emerged as a popular robust alternative. It generalizes the continuous ranked probability score (CRPS, Matheson & Winkler, 1976; Gneiting & Raftery, 2005) for univariate distributions and effectively mitigates sensitivity to outliers by evaluating forecasts through expected pairwise distances between predictions and observations. However, the ES lacks a closed-form analytical expression in most cases, necessitating computationally intensive Monte Carlo sampling to approximate its value and gradients. Such approximations significantly slow down neural network training, limiting practical scalability (Pacchiardi et al., 2024; Chen et al., 2024).

Motivated by the need for a robust yet computationally efficient scoring rule, this paper introduces MVG-CRPS (Multivariate Gaussian CRPS). We propose a strictly proper scoring rule specifically designed for multivariate Gaussian probabilistic forecasting tasks. Our approach circumvents the

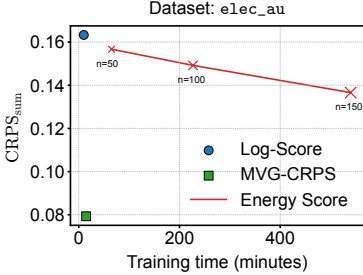

Figure 1: An example showing MVG-CRPS achieves better accuracy and faster training by avoiding sampling and reducing sensitivity to outliers. ES results are shown for different sample sizes.

computational limitations of the ES by leveraging a PCA whitening transformation, decomposing the multivariate Gaussian distribution into independent, standard normal variables. Consequently, the multivariate scoring problem reduces to a set of analytically tractable univariate CRPS computations. MVG-CRPS provides explicit analytical gradients, enabling efficient integration into neural network training. The advantages of our approach are illustrated in Fig. 1, where the model trained with MVG-CRPS achieves higher accuracy while significantly reducing training time. The key contributions of our work are:

- We propose MVG-CRPS, a novel scoring rule for multivariate probabilistic forecasting that is less sensitive to outliers and extreme tails of the data distribution. Under the multivariate Gaussian family, we prove that MVG-CRPS is strictly proper.
- The proposed MVG-CRPS has a closed-form expression, allowing for the analytical computation of derivatives. This property facilitates efficient integration with backpropagation-based training in deep learning models and significantly reduces the computational cost compared to sampling-based alternatives.
- We perform extensive experiments with deep probabilistic forecasting models on real-world datasets. Our results demonstrate that MVG-CRPS balances accuracy and efficiency more effectively than standard scoring rules.

## 2 RELATED WORK

### 2.1 PROBABILISTIC FORECASTING

Probabilistic forecasting focuses on modeling the complete probability distribution of target variables rather than producing single-point estimates. This comprehensive approach is essential for quantifying uncertainty inherent in time series data, thereby enabling more informed risk assessment and decision-making. Probabilistic forecasting methods typically fall into two main categories: parametric methods (e.g., through probability density functions (PDFs)) and non-parametric methods (e.g., through quantile functions) (Benidis et al., 2022).

Non-parametric methods generally forecast specific quantiles of the target distribution, thus avoiding restrictive parametric assumptions. A prominent example is the MQ-RNN (Wen et al., 2017), which leverages a Seq2Seq recurrent neural network (RNN) architecture to forecast multiple quantiles simultaneously. These quantile forecasts offer a robust approximation of the underlying distribution, making them particularly effective for capturing asymmetric and heavy-tailed behaviors.

Parametric methods assume a predefined probability distribution—such as Gaussian or Poisson—and estimate its parameters using neural networks. The DeepAR model (Salinas et al., 2020), for instance, employs an RNN to capture hidden state transitions and predict Gaussian distribution parameters at each time step. GPVar (Salinas et al., 2019), its multivariate extension, incorporates a Gaussian copula to transform observations into Gaussian variables, thus modeling joint dependencies among multiple time series effectively. This method efficiently captures temporal and cross-series correlations through generalized least squares (GLS) approaches (Zheng et al., 2024; Zheng & Sun, 2024) or dynamic regression (Zheng et al., 2025).

Neural networks also facilitate modeling more complex probabilistic structures, including state-space models (SSMs) (Rangapuram et al., 2018; de Bézenac et al., 2020), normalizing flows (NFs) (Rasul et al., 2021b), and diffusion models (Rasul et al., 2021a). Additionally, copula-based methods explicitly model dependencies between multiple time series. Recent studies by Drouin et al. (2022) and Ashok et al. (2024) employ copulas to combine individual marginal distributions and dependency structures, achieving flexible multivariate modeling capabilities. Most existing approaches predominantly use the log-score as their optimization criterion.

## 2.2 SCORING RULES

Scoring rules quantitatively assess probabilistic forecast quality by comparing predictive distributions with observed outcomes. A scoring rule assigns a number $S(F, z)$ to a predictive distribution $F$ and an observation $z$. For a true distribution $G$, define the expected score $S(F, G) = \mathbb{E}_{Z \sim G}\big[S(F, Z)\big]$. $S$ is *proper* if $S(G, G) \leq S(F, G)$ for all $F, G$, and *strictly proper* if equality implies $F = G$ (Bröcker, 2009).

The negative log-likelihood (log-score) is a prevalent strictly proper scoring rule, evaluating predictive densities directly at observed outcomes. Widely adopted due to its analytical tractability, the log-score is particularly beneficial when the predictive density has a known parametric form (Panagiotelis et al., 2023). The log-score is a strictly proper scoring rule and has several desirable properties, such as consistency and sensitivity to the entire distribution. In addition, the analytical tractability (closed-form expression and gradients for many distributions) makes it a convenient default in deep probabilistic forecasting models. However, for certain distributions (e.g., Gaussian), the log-score severely penalizes unlikely events, rendering it sensitive to outliers and extreme observations (Gneiting et al., 2007). To mitigate this sensitivity, the CRPS provides a robust alternative in univariate contexts (Rasp & Lerch, 2018). The CRPS quantifies discrepancies between the predictive cumulative distribution function (CDF) and observations, integrating absolute error over all potential thresholds. Unlike the exponential penalty in log-score, CRPS linearly penalizes deviations, thus reducing vulnerability to extreme events (Gneiting et al., 2005). CRPS-based optimization techniques have demonstrated superior calibration and robustness compared to likelihood-based approaches in various probabilistic forecasting applications (Gneiting et al., 2005; Olivares et al., 2023; Lang et al., 2024). Minimum CRPS estimation specifically targets improved calibration by optimizing parameters directly to minimize CRPS rather than maximizing likelihood.

Multivariate forecasting introduces additional complexity due to inter-dependencies and higher dimensionality. While the log-score remains applicable, its sensitivity to outliers persists in this setting. The ES (Gneiting & Raftery, 2007) generalizes the CRPS for multivariate distributions by computing expected distances between predictive and observed distributions. While ES effectively detects errors in the forecast mean, it is less sensitive to variance errors and, more critically, to misspecifications in the correlation structure among variables (Pinson & Tastu, 2013; Alexander et al., 2024). The absence of a closed form expression also necessitates the use of Monte Carlo simulations to approximate the ES by drawing samples from the predictive distribution, which can be computationally expensive (see e.g., Bouchacourt et al., 2016; Panagiotelis et al., 2023; Chen et al., 2024; Pacchiardi et al., 2024).

To overcome the limited sensitivity of ES to the dependence structure, the variogram score (VS) was proposed by Scheuerer & Hamill (2015). VS explicitly targets inter-variable dependencies by comparing pairwise differences between forecasted and observed components. Similar to the ES, VS is typically approximated using ensemble forecasts or Monte Carlo sampling. However, it introduces additional computational complexity and still lacks a fully closed-form expression, limiting its direct applicability in large-scale or real-time settings. For a broader discussion of multivariate scoring rules and their properties, we refer readers to the comprehensive reviews by Gneiting & Katzfuss (2014), Ziel & Berk (2019), Waghmare & Ziegel (2025) and Pic et al. (2025).

The most relevant recent work is by Olafsdottir et al. (2024), who propose a parameter estimation framework for multivariate spatial models by maximizing the average leave-one-out score (LOOS). Their method leverages the tractable conditionals of multivariate Gaussians and robust scoring rules like the CRPS. It is especially efficient for models with sparse precision matrices (e.g., Gaussian Markov random fields), but incurs notable overhead for general multivariate Gaussians due to the cost of computing all conditionals.

## 3 OUR METHOD

### 3.1 MULTIVARIATE PROBABILISTIC FORECASTING

Probabilistic forecasting aims to estimate the joint conditional distribution over a collection of future quantities based on a given history of observations (Gneiting & Katzfuss, 2014). Denote the values of time series at time $t$ as $\mathbf{z}_t = [z_{1,t}, \ldots, z_{N,t}]^\top \in \mathbb{R}^N$, where $N$ is the number of series. The problem of probabilistic forecasting can be formulated as $p(\mathbf{z}_{T+1:T+Q} \mid \mathbf{z}_{T-P+1:T}; \mathbf{x}_{T-P+1:T+Q})$, where $\mathbf{z}_{t_1:t_2} = [\mathbf{z}_{t_1}, \ldots, \mathbf{z}_{t_2}]$, $P$ is the conditioning range, $Q$ is the prediction range, and $T$ is the time point that splits the conditioning range and prediction range. $\mathbf{x}_t$ are some known covariates for both past and future time steps.

Multivariate probabilistic forecasting can be formulated in different ways. One way is over the time series dimension, where multiple interrelated variables are forecast simultaneously at each time point. Considering an autoregressive model, where the predicted output is used as input for the next time step, this formulation can be factorized as

$$p(\mathbf{z}_{T+1:T+Q} \mid \mathbf{z}_{T-P+1:T}; \mathbf{x}_{T-P+1:T+Q}) = \prod_{t=T+1}^{T+Q} p(\mathbf{z}_t \mid \mathbf{z}_{t-P:t-1}; \mathbf{x}_{t-P:t}) = \prod_{t=T+1}^{T+Q} p(\mathbf{z}_t \mid \mathbf{h}_t),$$
(1)

where $\mathbf{h}_t$ is a state vector that encodes all the conditioning information used to generate the distribution parameters, typically via a neural network.

Another option is over the prediction horizon, where forecasts are made across multiple future time steps for one or more variables, capturing temporal dependencies and uncertainties over time. Considering a shared model across different series:

$$p(\mathbf{z}_{i,T+1:T+Q} \mid \mathbf{z}_{i,T-P+1:T}; \mathbf{x}_{i,T-P+1:T+Q}),$$
(2)

where $i = 1, \ldots, N$ denotes the identifier of a particular time series. Since the model outputs forecasts for the entire prediction horizon directly, it is also called a Seq2Seq model. Without loss of generality, we use the first approach as an example to illustrate our method, since both approaches focus on estimating a multivariate distribution $p(\mathbf{z}_t)$ or $p(\mathbf{z}_{i,T+1:T+Q})$.

A typical probabilistic forecasting model assumes Gaussian noise; for example, we can parameterize the joint distribution as

$$\mathbf{Z}_t \mid \mathbf{H}_t = \mathbf{h}_t \sim \mathcal{N}(\boldsymbol{\mu}(\mathbf{h}_t), \boldsymbol{\Sigma}(\mathbf{h}_t)),$$
(3)

where $\boldsymbol{\mu}(\cdot)$ and $\boldsymbol{\Sigma}(\cdot)$ are the functions mapping $\mathbf{h}_t$ to the mean and covariance parameters. The log-likelihood of the distribution given observed time series data up to time point $T$ can be used as the loss function for optimizing a deep learning model:

$$\mathcal{L} = \sum_{t=1}^{T} \log p(\mathbf{z}_t \mid \theta(\mathbf{h}_t)) \propto \sum_{t=1}^{T} -\frac{1}{2}[\ln|\boldsymbol{\Sigma}_t| + \boldsymbol{\eta}_t^\top \boldsymbol{\Sigma}_t^{-1} \boldsymbol{\eta}_t],$$
(4)

where $\boldsymbol{\eta}_t = \mathbf{z}_t - \boldsymbol{\mu}_t$. The above formulation simplifies to the univariate case when we set $N = 1$ for the model, with the same model being shared across all time series:

$$Z_{i,t} \mid \mathbf{H}_{i,t} = \mathbf{h}_{i,t} \sim \mathcal{N}(\mu(\mathbf{h}_{i,t}), \sigma^2(\mathbf{h}_{i,t})),$$
(5)

where $\mu(\cdot)$ and $\sigma(\cdot)$ map $\mathbf{h}_{i,t}$ to the mean and standard deviation of a Gaussian distribution. The corresponding log-likelihood becomes

$$\mathcal{L} = \sum_{t=1}^{T} \sum_{i=1}^{N} \log p(z_{i,t} \mid \theta(\mathbf{h}_{i,t})) \propto \sum_{t=1}^{T} \sum_{i=1}^{N} -\frac{1}{2}\epsilon_{i,t}^2 - \ln \sigma_{i,t},$$
(6)

where $\epsilon_{i,t} = \frac{z_{i,t} - \mu_{i,t}}{\sigma_{i,t}}$. Eq. (4) and Eq. (6), when used as scoring rules to optimize the model, are generally referred to as log-scores under Gaussian assumption and are widely employed in probabilistic forecasting.

For univariate problems, the CRPS is also a strictly proper scoring rule, defined as

$$\mathrm{CRPS}(F, z) = \mathop{\mathbb{E}}_{X \sim F}|X - z| - \frac{1}{2}\mathop{\mathbb{E}}_{X, X' \sim F}|X - X'|,$$
(7)

where $F$ is the predictive CDF, $z$ is the observation, and $X$ and $X'$ are independent random variables both associated with $F$. The closed-form expression of CRPS for Gaussian distribution can be derived as (Gneiting et al., 2005):

$$\text{CRPS}\left(\Phi, z\right) = z\left(2\Phi\left(z\right) - 1\right) + 2\varphi\left(z\right) - \frac{1}{\sqrt{\pi}}, \tag{8}$$

$$\text{CRPS}\left(F_{\mu,\sigma}, z\right) = \sigma\,\text{CRPS}\left(\Phi, \frac{z-\mu}{\sigma}\right), \tag{9}$$

where $F_{\mu,\sigma}\left(z\right) = \Phi\left(\frac{z-\mu}{\sigma}\right)$, $\Phi$ and $\varphi$ are the CDF and PDF of the standard Gaussian distribution.

CRPS has been shown to be a more robust alternative to the log-score as a loss function in many problems (Gneiting et al., 2005; Rasp & Lerch, 2018; Murad et al., 2021). We observe that the log-score can grow arbitrarily large in magnitude when a single outlier disproportionately influences the loss function, owing to the unbounded nature of the logarithmic function (Eq. (4) and Eq. (6)). Additionally, the quadratic form of the error terms in the Gaussian likelihood also makes it sensitive to outliers (e.g., $\epsilon_{i,t}^2$ in Eq. (6)). In contrast, the CRPS evaluates the entire predictive distribution rather than concentrating solely on the likelihood of individual data points (Eq. (8)). Moreover, the CRPS can directly replace the log-score, providing analytical gradients with respect to $\mu$ and $\sigma$ for backpropagation. However, for a multivariate Gaussian distribution, the CRPS does not have a widely used closed-form expression.

## 3.2 MVG-CRPS AS LOSS FUNCTION FOR MULTIVARIATE FORECASTING

In multivariate probabilistic forecasting, proper scoring rules such as the log-score (Eq. (4)) and the ES are used to evaluate predictive performance. The ES generalizes the CRPS to assess probabilistic forecasts of vector-valued random variables (Gneiting & Raftery, 2007):

$$\text{ES}(F, \mathbf{z}) = \mathop{\mathbb{E}}_{\boldsymbol{X} \sim F} \|\boldsymbol{X} - \mathbf{z}\|^{\beta} - \frac{1}{2} \mathop{\mathbb{E}}_{\boldsymbol{X}, \boldsymbol{X}' \sim F} \|\boldsymbol{X} - \boldsymbol{X}'\|^{\beta}, \tag{10}$$

where $\|\cdot\|$ denotes the Euclidean norm and $\beta \in (0, 2)$, for which ES is strictly proper. With $\beta = 1$, the ES essentially becomes a multivariate extension of the CRPS and grows linearly with respect to the norm, making it less sensitive to outliers compared to the log-score. Since there is no simple closed-form expression for Eq. (10), it is often approximated using Monte Carlo methods, where multiple samples $\{\boldsymbol{x}_i\}_{i=1}^n$ are drawn from the forecast distribution to approximate the expected values:

$$\widehat{\text{ES}}(F, \mathbf{z}) = \frac{1}{n}\sum_{i=1}^{n}\|\boldsymbol{x}_i - \mathbf{z}\|^{\beta} - \frac{1}{2n^2}\sum_{i=1}^{n}\sum_{j=1}^{n}\|\boldsymbol{x}_i - \boldsymbol{x}_j\|^{\beta}. \tag{11}$$

However, a significant disadvantage of using Eq. (11) as the loss function is that it requires Monte Carlo sampling during the training process, which can substantially slow down training and create noisy gradients.

In this section, we propose the MVG-CRPS, a robust and efficient loss function designed as an alternative for multivariate forecasting. This loss function grows linearly with the prediction error, making it more robust than the log-score. Additionally, it does not require sampling during the training process, rendering it more efficient than the ES.

Our proposed method is based on the whitening transformation of a time series vector that follows a multivariate Gaussian distribution, $\mathbf{Z}_t \sim \mathcal{N}\left(\boldsymbol{\mu}_t, \boldsymbol{\Sigma}_t\right)$ (here we omit the conditioning for simplicity). The whitening process transforms a random vector with a known covariance matrix into a new random vector whose covariance matrix is the identity matrix. As a result, the elements of the transformed vector have unit variance and are uncorrelated. This transformation begins by performing the singular value decomposition (SVD) of the covariance matrix:

$$\boldsymbol{\Sigma}_t = \boldsymbol{U}_t \boldsymbol{S}_t \boldsymbol{U}_t^{\top}, \tag{12}$$

where $\boldsymbol{S}_t = \text{diag}([\lambda_{1,t}, \ldots, \lambda_{N,t}]^{\top})$ is a diagonal matrix containing the eigenvalues of $\boldsymbol{\Sigma}_t$, and $\boldsymbol{U}_t$ is the orthonormal matrix of corresponding eigenvectors. We then define

$$\mathbf{v}_t = \boldsymbol{U}_t^{\top}\left(\mathbf{z}_t - \boldsymbol{\mu}_t\right), \tag{13}$$

where $\mathbf{v}_t$ is a random vector with a uncorrelated multivariate Gaussian distribution $\mathcal{N}\left(\mathbf{0}, \boldsymbol{S}_t\right)$, having variances $\lambda_i$ (i.e., the corresponding eigenvalue) along the diagonal of its covariance matrix. Next, we define

$$\mathbf{w}_t = \boldsymbol{S}_t^{-\frac{1}{2}} \mathbf{v}_t = \boldsymbol{S}_t^{-\frac{1}{2}} \boldsymbol{U}_t^\top \left(\mathbf{z}_t - \boldsymbol{\mu}_t\right), \tag{14}$$

where $\mathbf{w}_t$ is a random vector with each element $w_{i,t}$ following a standard Gaussian distribution $\mathcal{N}(0,1)$. We can then apply Eq. (8) individually to each element and formulate the MVG-CRPS mimicking Eq. (9) for multivariate problem:

$$\mathrm{MCRPS}\left(\Phi_N\left(\boldsymbol{\mu}_t, \boldsymbol{\Sigma}_t\right), \mathbf{z}_t\right) = \sum_{i=1}^{N} \mathrm{CRPS}\left(\Phi\left(0, \lambda_{i,t}\right), v_{i,t}\right) = \sum_{i=1}^{N} \sqrt{\lambda_{i,t}}\, \mathrm{CRPS}\left(\Phi, w_{i,t}\right), \tag{15}$$

where $\Phi_N(\boldsymbol{\mu}, \boldsymbol{\Sigma})$ is thd CDF of multivariate Gaussian with mean $\boldsymbol{\mu}$ and covariance $\boldsymbol{\Sigma}$.

The overall loss function for training the model is then formulated over an observation period $T$:

$$\mathcal{L} = \sum_{t=1}^{T} \mathrm{MCRPS}\left(\Phi_N\left(\boldsymbol{\mu}_t, \boldsymbol{\Sigma}_t\right), \mathbf{z}_t\right). \tag{16}$$

The key advantage of MVG-CRPS lies in its ability to exploit the closed-form expression of the univariate CRPS by decorrelating multivariate time series variables via PCA whitening. This transformation enables the evaluation of marginal distributions in an orthogonalized space, where the whitening is derived from the original covariance matrix. As a result, the optimization process preserves and is sensitive to the dependence structure of the original multivariate distribution. Under the Gaussian assumption, MVG-CRPS constitutes a strictly proper scoring rule (see Appendix §A).

## 4 EXPERIMENTS

### 4.1 SIMULATION STUDY

We first perform a toy experiment following Roordink & Hess (2023) using a true distribution $P = \mathcal{N}\left(\begin{bmatrix} 1 \\ -1 \end{bmatrix}, \begin{bmatrix} 1 & 0.8 \\ 0.8 & 4 \end{bmatrix}\right)$ and a predictive distribution $Q = \mathcal{N}\left(\begin{bmatrix} \mu \\ -1 \end{bmatrix}, \begin{bmatrix} \sigma^2 & 2\rho\sigma \\ 2\rho\sigma & 4 \end{bmatrix}\right)$, where we control the deviation of the three parameters $\mu, \rho, \sigma$ to study the various properties of different scores. As shown in Fig. 2, the log-score increases sharply when the standard deviation $\sigma$ or correlation coefficient $\rho$ deviate from their true values, indicating high sensitivity to covariance misspecification. The ES shows lower sensitivity to the covariance structure but produces non-smooth curves due to its sample-based approximation. In contrast, the MVG-CRPS displays comparable sensitivity to deviations in all three parameters. It also produces smooth curves with a clear minimum at zero deviation, reflecting its closed-form evaluation.

We further examine the robustness of different scoring rules for estimating the parameters of this predictive distribution under data contamination, and analyze the trade-off between computational cost and estimation accuracy for the ES with varying sample sizes (see Appendix §D.1). Overall, MVG-CRPS demonstrates greater robustness than the log-score across all three parameters, particularly for $\mu$ and $\sigma$, and provides more consistent estimates than the ES due to its sampling-free formulation (Fig. A1). We also observe that the ES produces less accurate estimates than MVG-CRPS for $\mu$ and $\sigma$. Although we do not claim superiority over the ES beyond efficiency, this discrepancy is likely attributable to the variance introduced by its Monte Carlo approximation. Additionally, we observe that the gains in estimation accuracy diminish rapidly as the sample size increases, and the ES does not significantly outperform MVG-CRPS even with 1,000 samples (Fig. A2). Meanwhile, the computational cost of the ES increases monotonically with sample size.

### 4.2 REAL DATA APPLICATION

We evaluate MVG-CRPS on two forecasting tasks: multivariate autoregressive forecasting using the RNN-based GPVar (Salinas et al., 2019) and a decoder-only Transformer (Radford et al., 2018), and univariate Seq2Seq forecasting using the MLP-based N-HiTS model (Challu et al., 2023).

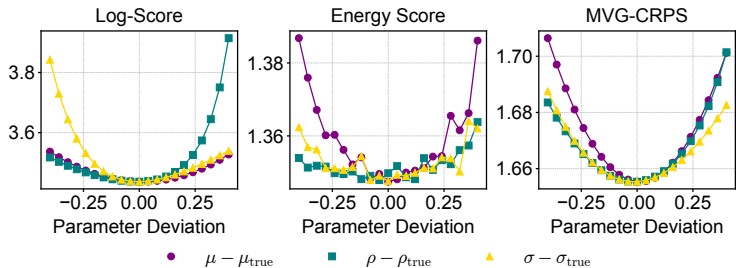

Figure 2: Sensitivity of scoring rules to parameter deviations in the predicted mean, standard deviation, and correlation coefficient from the true data distribution ($\mu_{\text{true}} = 1, \sigma_{\text{true}} = 1, \rho_{\text{true}} = 0.4$). The ES values are computed with a sample size of 500.

To generate the distribution parameters for probabilistic forecasting, we employ a Gaussian distribution head based on the hidden state $\mathbf{h}_{i,t}$ produced by the model. Specifically, for the multivariate autoregressive forecasting, following Salinas et al. (2019), we parameterize the mean vector as $\boldsymbol{\mu}(\mathbf{h}_t) = [\mu_1(\mathbf{h}_{1,t}), \ldots, \mu_N(\mathbf{h}_{N,t})]^\top \in \mathbb{R}^N$ and adopt a low-rank-plus-diagonal parameterization of the covariance matrix $\boldsymbol{\Sigma}(\mathbf{h}_t) = \boldsymbol{L}_t \boldsymbol{L}_t^\top + \text{diag}(\mathbf{d}_t)$, where $\mathbf{d}_t = [d_1(\mathbf{h}_{1,t}), \ldots, d_N(\mathbf{h}_{N,t})]^\top \in \mathbb{R}_+^N$ and $\boldsymbol{L}_t = [\mathbf{l}_1(\mathbf{h}_{1,t}), \ldots, \mathbf{l}_N(\mathbf{h}_{N,t})]^\top \in \mathbb{R}^{N \times R}$, $R \ll N$ is the rank parameter. Here, $\mu_i(\cdot)$, $d_i(\cdot)$, and $\mathbf{l}_i(\cdot)$ are the mapping functions that generate the mean and covariance parameters for each time series $i$ based on the hidden state $\mathbf{h}_{i=1:N,t}$. This parameterization guarantees that $\boldsymbol{\Sigma}(\mathbf{h}_t)$ is full-rank, ensuring that the eigen-decomposition in Eq. (12) is always well-defined. In practice, we use shared mapping functions across all time series, denoted as $\mu_i = \tilde{\mu}$, $d_i = \tilde{d}$, and $\mathbf{l}_i = \tilde{\mathbf{l}}$. This parameterization ensures that $\boldsymbol{\Sigma}(\mathbf{h}_t)$ is positive definite and efficiently parameterized. The diagonal component provides stability, while the low-rank component captures the covariance structure. The Gaussian assumption also enables the use of random subsets of time series (i.e., batch size $B \leq N$) for model optimization in each iteration, making it feasible to apply our method to high-dimensional time series datasets. Similarly, in the univariate Seq2Seq forecasting task, the mean $\boldsymbol{\mu}(\mathbf{h}_i)$ and covariance $\boldsymbol{\Sigma}(\mathbf{h}_i)$ are defined over the forecast horizon for each specific time series, based on the hidden states $\mathbf{h}_{i,t=T+1:T+Q}$. As a result, we can model the joint distribution $p(\mathbf{z}_{i,T+1:T+Q})$ over the forecasted values. We implemented our models using PyTorch Forecasting (Beitner, 2020), with input data consisting of lagged time series values and covariates. Extensive experiments were conducted on a variety of real-world time series datasets from GluonTS (Alexandrov et al., 2020) (see Appendix §B). Full details of the experimental setup are provided in Appendix §C.

### 4.2.1 QUANTITATIVE EVALUATION

We evaluate the MVG-CRPS against models trained with the log-score and the ES using three common metrics for probabilistic forecasts: $\text{CRPS}_{\text{sum}}$, $\text{CRPS}_{\text{mean}}$, and the ES (see Appendix §C.5 for definitions). Table 1 presents a comparison of $\text{CRPS}_{\text{sum}}$ for the multivariate autoregressive forecasting task. Overall, the MVG-CRPS achieves the best average rank among the three scoring rules. Notably, it consistently outperforms the log-score across most datasets, indicating that MVG-CRPS leads to models with higher-quality forecasts. As shown in later sections, this improvement is attributed to MVG-CRPS being less sensitive to outliers. Compared to the ES, MVG-CRPS achieves comparable or better performance (Table 1) while being more efficient during training (Table 2). It is important to note that we do not claim MVG-CRPS is more robust than ES; rather, our focus is on its efficiency compared to ES. Results for $\text{CRPS}_{\text{mean}}$ and the ES are provided in Appendix §D.2, and results for the univariate Seq2Seq forecasting task are presented in Appendix §D.3. In both tasks, MVG-CRPS achieves consistent performance across all three evaluation metrics.

### 4.2.2 QUALITATIVE EVALUATION

To illustrate the robustness of MVG-CRPS, we compare the output covariance matrices from models trained with different loss functions and visualize their probabilistic forecasts. As seen in Fig. 3, the log-score objective can induce occasional large covariance entries. ES and MVG-CRPS, however, do not exhibit these extremes. Importantly, MVG-CRPS retains a covariance pattern closer to the

Table 1: Comparison of $CRPS_{sum}$ across different scoring rules in the multivariate autoregressive forecasting task. The best scores are in boldface. MVG-CRPS scores are underlined when they are not the best overall but exceed the log-score.

| | VAR | GPVar | | | Transformer | | |
|---|---|---|---|---|---|---|---|
| | | log-score | energy score | MVG-CRPS | log-score | energy score | MVG-CRPS |
| elec_au | N/A | 0.1261±0.0009 | **0.0887±0.0004** | 0.0967±0.0008 | 0.1633±0.0005 | 0.1492±0.0006 | **0.0793±0.0004** |
| cif_2016 | 1.0000±0.0000 | 0.0122±0.0004 | 0.0420±0.0006 | **0.0111±0.0005** | 0.0118±0.0003 | 0.0240±0.0014 | **0.0107±0.0002** |
| electricity | 0.1315±0.0006 | 0.0419±0.0008 | 0.0616±0.0004 | **0.0249±0.0006** | 0.0362±0.0002 | 0.0368±0.0004 | **0.0294±0.0004** |
| elec_weekly | 0.1126±0.0011 | 0.1515±0.0028 | **0.0417±0.0014** | 0.0772±0.0031 | 0.0937±0.0026 | **0.0403±0.0013** | 0.0448±0.0014 |
| exchange_rate | 0.0033±0.0000 | 0.0207±0.0004 | **0.0030±0.0001** | 0.0041±0.0001 | **0.0047±0.0003** | 0.0067±0.0003 | 0.0091±0.0004 |
| kdd_cup | N/A | 0.3743±0.0019 | 0.3210±0.0019 | **0.2358±0.0014** | 0.2076±0.0013 | 0.4789±0.0030 | **0.1959±0.0017** |
| m1_yearly | N/A | 0.4397±0.0041 | 0.4801±0.0022 | **0.3566±0.0029** | 0.5344±0.0109 | **0.3291±0.0047** | 0.4563±0.0111 |
| m3_yearly | N/A | 0.3607±0.0084 | 0.2186±0.0042 | **0.1423±0.0053** | 0.3156±0.0102 | 0.4050±0.0061 | **0.2325±0.0094** |
| nn5_daily | 0.2303±0.0005 | 0.0998±0.0004 | 0.0958±0.0003 | **0.0948±0.0003** | 0.0991±0.0003 | 0.0883±0.0004 | **0.0811±0.0002** |
| saugeenday | N/A | 0.4040±0.0047 | **0.3733±0.0048** | 0.3941±0.0055 | 0.3771±0.0088 | **0.3689±0.0053** | 0.3705±0.0047 |
| sunspot | N/A | 18.7115±1.3296 | 23.3988±0.9662 | **17.2438±0.5833** | 39.7454±1.4841 | **16.6556±0.6167** | 22.6495±0.6752 |
| tourism | 0.1394±0.0012 | 0.2217±0.0027 | 0.2112±0.0014 | **0.2004±0.0022** | 0.2100±0.0017 | 0.2087±0.0020 | **0.2082±0.0015** |
| traffic | 3.5241±0.0084 | 0.0742±0.0004 | **0.0505±0.0002** | 0.0868±0.0002 | **0.0658±0.0002** | 0.0667±0.0002 | 0.0683±0.0000 |
| **Avg. Rank** | | 2.62 | 1.92 | **1.46** | 2.38 | 2.00 | **1.62** |

Table 2: Training time (in minutes) for GPVar using different scoring rules in the multivariate autoregressive forecasting task. Reported times include early stopping and reflect differences in convergence speed across loss functions.

| | log-score | | energy score | | MVG-CRPS | |
|---|---|---|---|---|---|---|
| | per epoch | total | per epoch | total | per epoch | total |
| elec_au | 0.86 | 33.53 | 16.29 | 717.00 | **0.78** | **29.14** |
| cif_2016 | 0.13 | **1.58** | 4.83 | 401.04 | **0.12** | 3.85 |
| electricity | 0.40 | 67.38 | 11.17 | 782.40 | **0.38** | **22.70** |
| elec_weekly | 0.30 | **14.61** | 10.95 | 383.52 | **0.26** | 18.77 |
| exchange_rate | **0.25** | **16.40** | 10.20 | 663.60 | 0.29 | 23.63 |
| kdd_cup | **0.42** | **11.32** | 14.23 | 2063.52 | **0.42** | 28.79 |
| m1_yearly | 0.19 | **3.71** | 5.66 | 469.92 | **0.18** | 8.02 |
| m3_yearly | 0.43 | **7.30** | 10.80 | 291.72 | **0.42** | 14.49 |
| nn5_daily | 0.29 | **9.21** | 11.64 | 244.50 | **0.27** | 14.53 |
| saugeenday | 0.23 | **12.65** | 10.70 | 524.46 | **0.15** | 15.32 |
| sunspot | 0.44 | 26.85 | 10.73 | 397.26 | **0.42** | **16.96** |
| tourism | 0.49 | 23.96 | 10.56 | 243.00 | **0.46** | **12.51** |
| traffic | 0.94 | **76.98** | 14.92 | 1044.60 | **0.92** | 92.46 |

log-score model, while controlling the magnitude of covariances. This aligns with the toy example in §4.1, which shows that ES is comparatively insensitive to covariance structure. To highlight the practical impact, we compare GPVar forecasts on the electricity dataset (Fig. 4). MVG-CRPS yields sharper and better-calibrated predictions, while the log-score model occasionally produces overly wide intervals, reflecting greater sensitivity to outliers (e.g., TS 1). Compared with ES, MVG-CRPS delivers comparable or better performance. Results for the univariate Seq2Seq forecasting task are provided in Appendix §D.3.

## 5 CONCLUSION

This paper introduced the MVG-CRPS, a novel strictly proper scoring rule specifically designed for multivariate Gaussian probabilistic forecasting. MVG-CRPS addresses the sensitivity of the log-score to outliers and overcomes the computational inefficiency inherent to the ES. By applying a whitening transformation and leveraging the closed-form expression of the univariate CRPS, our approach achieves robustness to extreme values while remaining computationally efficient and easily integrable into deep learning frameworks. Moreover, the MVG-CRPS exhibits high sensitivity to both the mean and covariance of the predictive distribution—comparable to the log-score—while preserving the robustness properties of the ES. Empirical evaluations on real-world datasets demonstrated significant improvements in both predictive accuracy and robustness compared to existing scoring rules.

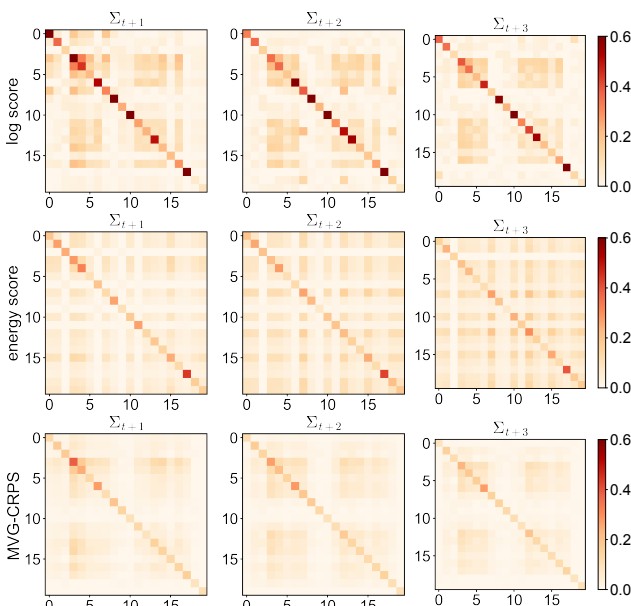

Figure 3: Comparison of output covariance matrices $\Sigma_t$ from GPVar on the `elec_weekly` dataset. For visual clarity, covariance values are clipped between 0 and 0.6.

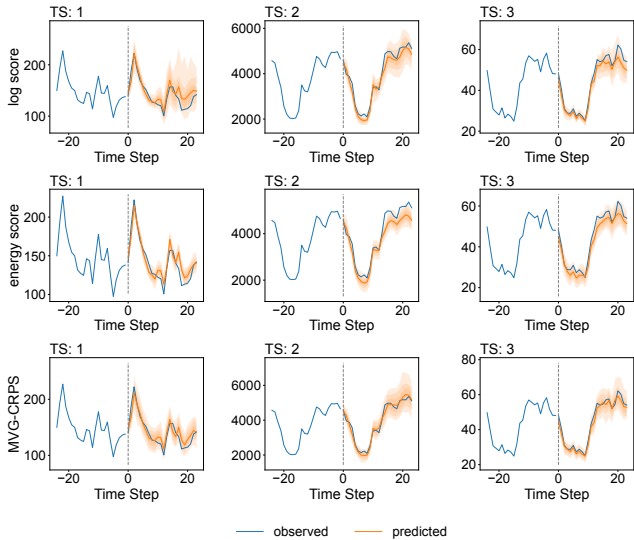

Figure 4: Comparison of probabilistic forecasts from GPVar on the `electricity` dataset.

Beyond forecasting, the general formulation of MVG-CRPS extends naturally to broader probabilistic regression contexts, such as robust Gaussian process regression, by replacing conventional negative marginal likelihood objectives. Future directions include leveraging copula transformations to extend the MVG-CRPS to non-Gaussian distributions and exploring more efficient covariance parameterizations to enhance scalability. Currently, scalability remains constrained by the computational demands of eigen-decomposition in large-batch scenarios. A possible solution to mitigate this limitation is to adopt an isotropic noise parameterization, i.e., $\Sigma = LL^\top + \sigma^2 I$, which enables more efficient computation of the SVD.

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

# Appendix

TABLE OF CONTENTS

## A  MVG-CRPS IS STRICTLY PROPER

**Theorem A.1.** *Let* $\mathbf{z} \sim \mathcal{N}\left(\boldsymbol{\mu}_p, \boldsymbol{\Sigma}_p\right)$ *be a true* $N$-*variate Gaussian distribution where the covariance admits eigen-decomposition* $\boldsymbol{\Sigma}_p = \boldsymbol{U}_p \boldsymbol{S}_p \boldsymbol{U}_p^\top$, *with* $\boldsymbol{S}_p = \mathrm{diag}\left(\boldsymbol{\lambda}_p\right)$ *containing nonincreasing eigenvalues* $\boldsymbol{\lambda}_p = [\lambda_1^p, \ldots, \lambda_N^p]^\top$ *and* $\boldsymbol{U}_p$ *being the corresponding orthonormal matrix. Consider a predictive Gaussian distribution* $\mathcal{N}\left(\boldsymbol{\mu}_q, \boldsymbol{\Sigma}_q\right)$, *where covariance* $\boldsymbol{\Sigma}_q$ *admits the eigen-decomposition* $\boldsymbol{\Sigma}_q = \boldsymbol{U}_q \boldsymbol{S}_q \boldsymbol{U}_q^\top$ *with* $\boldsymbol{S}_q = \mathrm{diag}\left(\boldsymbol{\lambda}_q\right)$. *Define the transformed variable* $\mathbf{v} = \boldsymbol{U}_q^\top \left(\mathbf{z} - \boldsymbol{\mu}_q\right) = [v_1, \ldots, v_N]^\top$. *The proposed MVG-CRPS*

$$\mathrm{MCRPS}\left(\Phi_N\left(\boldsymbol{\mu}_q, \boldsymbol{\Sigma}_q\right), \mathbf{z}\right) = \sum_{i=1}^{N} \mathrm{CRPS}\left(\Phi\left(0, \lambda_i^q\right), v_i\right)$$

*is proper and strictly proper for multivariate Gaussian distributions.*

*Proof.* Given that $\mathbf{z} \sim \mathcal{N}\left(\boldsymbol{\mu}_p, \boldsymbol{\Sigma}_p\right)$, we have the transformed variable $\mathbf{v} \sim \mathcal{N}\left(\boldsymbol{\mu}_v, \boldsymbol{\Sigma}_v\right)$ with $\boldsymbol{\mu}_v = \boldsymbol{U}_q^\top \left(\boldsymbol{\mu}_p - \boldsymbol{\mu}_q\right) = [\nu_1, \ldots, \nu_N]^\top$ and $\boldsymbol{\Sigma}_v = \boldsymbol{U}_q^\top \boldsymbol{\Sigma}_p \boldsymbol{U}_q = \boldsymbol{U}_q^\top \boldsymbol{U}_p \boldsymbol{S}_p \boldsymbol{U}_p^\top \boldsymbol{U}_q = \boldsymbol{U}_v \boldsymbol{S}_p \boldsymbol{U}_v^\top$, where $\boldsymbol{U}_v = \boldsymbol{U}_q^\top \boldsymbol{U}_p$ is an orthonormal matrix. Thus, each $v_i$ has a marginal distribution $v_i \sim \mathcal{N}\left(\nu_i, \tau_i\right)$ for $i = 1, \ldots, N$, with $\boldsymbol{\tau} = \mathrm{diag}(\boldsymbol{\Sigma}_v) = \mathrm{diag}(\boldsymbol{U}_v \boldsymbol{S}_p \boldsymbol{U}_v^\top) = [\tau_1, \ldots, \tau_N]^\top$. Taking the expectation of

$\text{MCRPS} \left( \Phi_N \left( \boldsymbol{\mu}_q, \boldsymbol{\Sigma}_q \right), \mathbf{z} \right)$ under the true distribution, we have

$$
\begin{aligned}
\mathbb{E}_{\mathbf{z} \sim \mathcal{N}(\boldsymbol{\mu}_p, \boldsymbol{\Sigma}_p)} \left[ \text{MCRPS} \left( \Phi_N \left( \boldsymbol{\mu}_q, \boldsymbol{\Sigma}_q \right), \mathbf{z} \right) \right] &= \sum_{i=1}^{N} \mathbb{E}_{v_i \sim \mathcal{N}(\nu_i, \tau_i)} \left[ \text{CRPS} \left( \Phi \left( 0, \lambda_i^q \right), v_i \right) \right] \\
&\geq \sum_{i=1}^{N} \mathbb{E}_{v_i \sim \mathcal{N}(\nu_i, \tau_i)} \left[ \text{CRPS} \left( \Phi \left( \nu_i, \tau_i \right), v_i \right) \right] \\
&= \sum_{i=1}^{N} \mathbb{E}_{\eta_i \sim \mathcal{N}(0, \tau_i)} \left[ \text{CRPS} \left( \Phi \left( 0, \tau_i \right), \eta_i \right) \right] \quad (17) \\
&= \mathbb{E}_{v \sim \mathcal{N}(0,1)} \left[ \text{CRPS} \left( \Phi, v \right) \right] \times \sum_{i=1}^{N} \sqrt{\tau_i} \\
&\geq \mathbb{E}_{v \sim \mathcal{N}(0,1)} \left[ \text{CRPS} \left( \Phi, v \right) \right] \times \sum_{i=1}^{N} \sqrt{\lambda_i^p} \\
&= \mathbb{E}_{\mathbf{z} \sim \mathcal{N}(\boldsymbol{\mu}_p, \boldsymbol{\Sigma}_p)} \left[ \text{MCRPS} \left( \Phi_N \left( \boldsymbol{\mu}_p, \boldsymbol{\Sigma}_p \right), \mathbf{z} \right) \right].
\end{aligned}
$$

The first inequality is a direct result of CRPS being a strictly proper scoring rule for univariate Gaussian distributions. We now prove the second inequality.

Recall that $\boldsymbol{\tau} = \text{diag}(\boldsymbol{\Sigma}_v)$ and $\boldsymbol{\Sigma}_v = \boldsymbol{U}_q^\top \boldsymbol{\Sigma}_p \boldsymbol{U}_q$. Let $\boldsymbol{\tau}^*$ be the monotone nonincreasing rearrangement of $\boldsymbol{\tau}$. By the Schur-Horn theorem Horn (1954), the diagonal vector $\boldsymbol{\tau}^*$ is majorized by the eigenvalues $\boldsymbol{\lambda}_p$:

$$
\sum_{i=1}^{k} \tau_i^* \leq \sum_{i=1}^{k} \lambda_i^p,
$$

for $k = 1, 2, \ldots, N-1$, and

$$
\sum_{i=1}^{N} \tau_i^* = \sum_{i=1}^{N} \lambda_i^p.
$$

Since $f(x) = \sqrt{x}$ is a concave function, Karamata's majorization inequality yields

$$
\sum_{i=1}^{N} \sqrt{\lambda_i^p} \leq \sum_{i=1}^{N} \sqrt{\tau_i^*} = \sum_{i=1}^{N} \sqrt{\tau_i}, \quad (18)
$$

which proves the second inequality in Eq. (17). Hence, the MVG-CRPS is a proper scoring rule for the multivariate Gaussian distribution.

Equality in Eq. (18) is obtained if, for every $i$, $\tau_i^* = \lambda_i^p$. By the Schur-Horn theorem, this forces $\boldsymbol{\Sigma}_v$ to be a diagonal matrix (Theorem 4.3.45 in Horn & Johnson (2012)). Meanwhile, the CRPS inequality in Eq. (17) is tight exactly when, for every $i$, $\nu_i = 0$ and $\tau_i = \lambda_i^q$, implying that $\boldsymbol{U}_q^\top \left( \boldsymbol{\mu}_p - \boldsymbol{\mu}_q \right) = \mathbf{0}$ and $\text{diag}(\boldsymbol{\Sigma}_v) = \text{diag}(\boldsymbol{S}_q)$. Since $\boldsymbol{\Sigma}_v$ is diagonal, we have $\boldsymbol{\Sigma}_v = \boldsymbol{U}_q^\top \boldsymbol{\Sigma}_p \boldsymbol{U}_q = \boldsymbol{S}_q$, hence $\boldsymbol{\Sigma}_p = \boldsymbol{\Sigma}_q$. Therefore, all equalities hold if and only if $\boldsymbol{\mu}_p = \boldsymbol{\mu}_q$ and $\boldsymbol{\Sigma}_p = \boldsymbol{\Sigma}_q$. This confirms that the proposed scoring rule is proper and strictly proper for the multivariate Gaussian distribution. $\square$

## B   DATASET DETAILS

We conducted experiments on a diverse collection of real-world datasets sourced from GluonTS (Alexandrov et al., 2020). These datasets are commonly used for benchmarking time series forecasting models, following their default configurations in GluonTS, which include granularity, prediction horizon ($Q$), and the number of rolling evaluations. For each dataset, we sequentially split the data into training, validation, and testing sets, ensuring that the temporal length of the validation set matched that of the testing set. The temporal length of the testing set was based on the prediction horizon and the required number of rolling evaluations. For example, the testing horizon for the `traffic` dataset is calculated as $24 + 7 - 1 = 30$ time steps. Consequently, the model generates 24-step predictions ($Q$) sequentially, with 7 distinct consecutive prediction start points, corresponding

to 7 forecast instances. In our experiments, we aligned the conditioning range ($P$) with the prediction horizon ($Q$), consistent with the default setting in GluonTS (i.e., $P = Q$). Each time series was individually normalized using a scaler fitted to its own training data (Salinas et al., 2020; Kim et al., 2021). Predictions were then rescaled to their original values for computing evaluation metrics. Table A1 summarizes the statistics of all datasets.

Table A1: Dataset summary.

| Dataset | Granularity | # of time series | # of time steps | $Q$ | Rolling evaluation |
|---|---|---|---|---|---|
| elec_au | 30min | 5 | 232,272 | 60 | 56 |
| cif_2016 | monthly | 72 | 120 | 12 | 1 |
| electricity | hourly | 370 | 5,857 | 24 | 7 |
| elec_weekly | weekly | 321 | 156 | 8 | 3 |
| exchange_rate | workday | 8 | 6,101 | 30 | 5 |
| kdd_cup | hourly | 270 | 10,920 | 48 | 7 |
| m1_yearly | yearly | 181 | 169 | 6 | 1 |
| m3_yearly | yearly | 645 | 191 | 6 | 1 |
| nn5_daily | daily | 111 | 791 | 56 | 5 |
| saugeenday | daily | 1 | 23,741 | 30 | 5 |
| sunspot | daily | 1 | 73,924 | 30 | 5 |
| tourism | quarterly | 427 | 131 | 8 | 1 |
| traffic | hourly | 963 | 4,025 | 24 | 7 |
| covid | daily | 266 | 212 | 30 | 5 |
| elec_hourly | hourly | 321 | 26,304 | 48 | 7 |
| m4_hourly | hourly | 414 | 1,008 | 48 | 7 |
| pedestrian | hourly | 66 | 96,432 | 48 | 7 |
| taxi_30min | 30min | 1214 | 1,637 | 24 | 56 |
| uber_hourly | hourly | 262 | 8,343 | 24 | 7 |
| wiki | daily | 2000 | 792 | 30 | 5 |

## C EXPERIMENT DETAILS

### C.1 BENCHMARK MODELS

The input to benchmark models includes lagged time series values and covariates that encode time and series identification. The number of lagged values is determined by the granularity of each dataset. Specifically, we use lags of $\{1, 24, 168\}$ for hourly data, $\{1, 7, 14\}$ for daily data, and $\{1, 2, 4, 12, 24, 48\}$ for data with sub-hourly granularity. For all other datasets, only lag-1 values are used.

For datasets with hourly or finer granularity, we include the hour of the day and day of the week. For daily datasets, only the day of the week is used. Each time series is uniquely identified by a numeric identifier. All features are encoded as single values; for example, the hour of the day takes values between $[0, 23]$. These features are concatenated with the model input at each time step to form the model input vector $\mathbf{y}_t$ (Salinas et al., 2019; Zheng & Sun, 2024).

Our method requires a state vector $\mathbf{h}_{i,t}$ to generate the parameters for the predictive distribution. To achieve this, we employ different neural architectures: RNNs and Transformer decoders, both of which maintain autoregressive properties for the multivariate autoregressive forecasting task, and MLPs for the univariate Seq2Seq forecasting task. Specifically, we use the GPVar model (Salinas et al., 2019) as our RNN benchmark, the GPT model (Radford et al., 2018) for the decoder-only Transformer, and the N-HiTS model (Challu et al., 2023) for the MLPs. All models are trained to output $\mathbf{h}_{i,t}$, which is used to parameterize the predictive distribution.

### C.2 NAIVE BASELINE DESCRIPTION

In this paper, we use Vector Autoregression (VAR) (Lütkepohl, 2005) as a naive baseline model. The VAR($p$) model is formulated as

$$\mathbf{z}_t = \mathbf{c} + \boldsymbol{A}_1 \mathbf{z}_{t-1} + \cdots + \boldsymbol{A}_p \mathbf{z}_{t-p} + \boldsymbol{\epsilon}_t, \quad \boldsymbol{\epsilon}_t \sim \mathcal{N}(\mathbf{0}, \boldsymbol{\Sigma}_\epsilon), \tag{19}$$

where $A_i$ is an $N \times N$ coefficient matrix, and $\mathbf{c}$ is the intercept term. In our experiments, we employ a VAR model with a lag of 1 (VAR(1)). The parameters in Eq. (19) are estimated using ordinary least squares (OLS), as described in Lütkepohl (2005). VAR models are not applied to datasets with insufficient time series in the testing set and are marked as "N/A" in this paper.

## C.3 HYPERPARAMETERS

All model parameters are optimized using the Adam optimizer with $l_2$ regularization set to $1e^{-8}$, and gradient clipping applied at 10.0. For all methods, we cap the total number of gradient updates at 10,000 and reduce the learning rate by a factor of 2 after 500 consecutive updates without improvement. Table A2 provides the hyperparameter values that remain fixed across all datasets. In the main manuscript, we do **NOT** tune the hyperparameters specifically to favor the proposed loss. Instead, we use the same hyperparameters as those in GPVar (Salinas et al., 2019), which were originally tuned for the log-score. Keeping the hyperparameters consistent across loss functions ensures that any observed improvements are attributable to the loss function itself rather than differences in hyperparameter settings. However, we conduct additional studies using hyperparameters tuned for each loss function in §D.4.

Table A2: Hyperparameters values.

| Hyperparameter | Value |
|---|---|
| learning rate | 1e-3 |
| hidden size | 40 |
| n_layers (RNN/Transformer decoder/MLP) | 2 |
| n_heads (Transformer) | 2 |
| rank ($R$) | 10 |
| sampling dimension ($B$) | 20 |
| dropout | 0.01 |
| batch size | 16 |

## C.4 TRAINING PROCEDURE

**Compute Resources** All models were trained in an Anaconda environment using one AMD Ryzen Threadripper PRO 5955WX CPU and four NVIDIA RTX A5000 GPUs, each with 24 GB of memory.

**Batch Size** Following the method used in GPVar (Salinas et al., 2019), we set the sample slice size to $B = 20$ time series and used a batch size of 16. Since our data sampler processes one slice of time series at a time rather than sampling 16 slices simultaneously, we set `accumulate_grad_batches` to 16, effectively achieving a batch size of 16.

**Training Loop** During each epoch, the model is trained on up to 400 batches from the training set, followed by the computation of the `valid_loss` on the validation set. Training is halted when one of the following conditions is met:

- A total of 10,000 gradient updates has been reached,
- No improvement in the validation set `valid_loss` is observed for 10 consecutive epochs.

The final model is the one that achieves the lowest `valid_loss` on the validation set.

**Covariance Parameterization** The covariance matrix $\mathbf{\Sigma}_t$ is parameterized directly by the forecasting model. Specifically, it is constructed as: $\mathbf{\Sigma}_t = \mathbf{L}_t \mathbf{L}_t^\top + \text{diag}(\mathbf{d}_t)$, where $\mathbf{L}_t$ is a low-rank matrix and $\mathbf{d}_t$ is a positive diagonal vector. This parameterization ensures that $\mathbf{\Sigma}_t$ remains positive semi-definite while being computationally efficient to learn. This parameterization is standard in probabilistic forecasting and allows the model to learn both the structure (through $\mathbf{L}_t$) and scale (through $\mathbf{d}_t$) of the covariance during training. Without constraints, the MVG-CRPS loss could potentially be minimized by driving all eigenvalues of $\mathbf{\Sigma}_t$ to zero, resulting in a trivial solution. However, this is prevented through the following mechanisms:

- The diagonal entries of the covariance matrix are parameterized as $d_{i,t} = \texttt{softplus}(d_{i,t} + \texttt{diag\_bias}) + \sigma_{\min}^2$, where the `softplus` function ensures that the diagonal entries are

strictly positive, regardless of the raw input values, `diag_bias` is initialized to approximately `softplus_inv`$(\sigma_{\text{init}}^2)$, ensuring that the diagonal entries are initially close to $\sigma_{\text{init}}^2$. For instance, with $\sigma_{\text{init}} = 1.0$, the initial diagonal values start near 1.0. The addition of $\sigma_{\text{min}}^2$ provides a lower bound on the diagonal entries, ensuring that eigenvalues cannot approach zero.

- The low-rank component is parameterized as $\mathbf{L}_{i,t} = \frac{\mathbf{L}_{i,t}}{\sqrt{R}}$, where dividing by rank ensures that the low-rank term is well-scaled relative to the diagonal entries. This normalization prevents the low-rank component from dominating or becoming disproportionately small in the covariance matrix.

Moreover, the MVG-CRPS loss provides a balance between the calibration and sharpness of the forecasts:

$$\mathbf{w}_t = \boldsymbol{S}_t^{-\frac{1}{2}} \mathbf{v}_t = \boldsymbol{S}_t^{-\frac{1}{2}} \boldsymbol{U}_t^\top (\mathbf{z}_t - \boldsymbol{\mu}_t),$$

$$\mathcal{L} = \sum_{t=1}^{T} \sum_{i=1}^{N} \sqrt{\lambda_t^i}\, \mathrm{CRPS}\,(\Phi, w_{i,t}).$$

We observe that if the eigenvalues $\lambda_t^i$ in $\boldsymbol{S}_t$ approach zero, $w_{i,t}$ will be scaled very aggressively. This leads to inflated residuals $w_{i,t}$, which subsequently affect the CRPS computation. Since the CRPS metric integrates over the forecast distribution $F(y)$, penalizing deviations between $F(y)$ and the empirical step function $\mathbf{1}(y \geq w_{i,t})$, artificially large $w_{i,t}$ values (resulting from extreme eigenvalue scaling) will cause the CRPS term to increase significantly. This behavior reflects the importance of ensuring that eigenvalues $\lambda_t^i$ are well-regularized to prevent distortion in the forecast evaluation. By balancing the eigenvalue contributions, the MVG-CRPS ensures both stable calibration and sharpness in probabilistic forecasting.

**SVD and Gradient Calculation** We perform SVD on $\boldsymbol{\Sigma}(\mathbf{h}_t)$ to obtain $\mathbf{U}_t$ and $\mathbf{S}_t$ (the eigenvectors and eigenvalues, respectively). These are required to compute the whitening transformation: $\mathbf{w}_t = \mathbf{S}_t^{-\frac{1}{2}} \mathbf{U}_t^\top (\mathbf{z}_t - \boldsymbol{\mu}_t)$. During training, gradients of $\mathcal{L}$ need to flow back through the whitened vecotr $\mathbf{w}_t$, the eigenvectors matrix $\mathbf{U}_t$, the eigenvalues matrix $\mathbf{S}_t$, and the covariance matrix $\boldsymbol{\Sigma}_t$. The gradient of $\mathcal{L}$ with respect to $\mathbf{w}_t$ is $\frac{\partial \mathcal{L}}{\partial \mathbf{w}_t}$. Gradients of $\mathbf{w}_t$ are propagated to the whitening transformation: $\mathbf{w}_t = \mathbf{S}_t^{-\frac{1}{2}} \mathbf{U}_t^\top (\mathbf{z}_t - \boldsymbol{\mu}_t)$, which involves: (1) gradients with respect to $\mathbf{U}_t$; (2) gradients with respect to $\mathbf{S}_t^{-\frac{1}{2}}$ (i.e., the square root and inverse of singular values); and (3) gradients with respect to $(\mathbf{z}_t - \boldsymbol{\mu}_t)$. Using PyTorch's `torch.linalg.svd`, we calculate the gradients of $\mathbf{U}_t$ and $\mathbf{S}_t$ via automatic differentiation. For the forward pass, the cost of SVD for $\boldsymbol{\Sigma}(\mathbf{h}_t) \in \mathbb{R}^{B \times B}$ is $O(B^3)$, where $B$ is the matrix dimension. For the backward pass, computing the gradients of $\mathbf{U}_t$ and $\mathbf{S}_t$ also incurs $O(B^3)$ computational cost. Memory usage scales as $O(B^2)$ for storing the covariance matrix and the singular value decomposition outputs $(\mathbf{U}_t, \mathbf{S}_t)$. Additional memory is required for autograd intermediate values, scaling as $O(B^3)$. By leveraging PyTorch's autograd system, we integrate the computation of $\mathbf{U}_t$, $\mathbf{S}_t$, and their gradients seamlessly into our end-to-end learning pipeline. This ensures that the whitening transformation and the loss function are fully differentiable, allowing the model parameters to be trained via gradient-based optimizers. The parameter $B$ also plays a crucial role in the scalability of our method. By leveraging the Gaussian assumption, we are able to train the model using a much smaller subset of time series at each step. Consequently, the size of the covariance matrix is reduced to $B \times B$, as opposed to $N \times N$, where $N$ represents the total number of time series in the dataset. This design ensures that the computational complexity of our method does not scale with $N$. Moreover, $B$ is kept relatively small in our implementation (e.g., $B = 20$), making the approach computationally efficient.

## C.5 EVALUATION METRICS

In this paper, we repeated the evaluation procedure on the testing set ten times to compute the mean and standard deviation of each metric. For each evaluation, the metrics were calculated by averaging over all forecast instances in the testing set. For example, the reported $\mathrm{CRPS}_{\text{sum}}$ represents the average $\mathrm{CRPS}_{\text{sum}}$ across all forecast instances. Both CRPS and ES were estimated using Monte Carlo approximation based on 100 sampled predictions.

### C.5.1 Continuous Ranked Probability Score

The empirical approximation of the Continuous Ranked Probability Score (CRPS) based on a finite sample $\{x_1, \ldots, x_n\}$ drawn from the predictive distribution $F$ is given by:

$$\widehat{\text{CRPS}}(F, z) = \frac{1}{n} \sum_{i=1}^{n} |x_i - z| - \frac{1}{2n^2} \sum_{i=1}^{n} \sum_{j=1}^{n} |x_i - x_j|, \tag{20}$$

where the first term estimates the expected absolute deviation between the predictive samples and the observation $z$, while the second term estimates the expected absolute deviation between pairs of predictive samples. This Monte Carlo approximation converges to the true CRPS as $n \to \infty$. An efficient empirical approximation of Eq. (20), based on a sorted sample $\{x_{(1)}, \ldots, x_{(n)}\}$ from the predictive distribution $F$, is given by:

$$\widehat{\text{CRPS}}(F, z) = \frac{1}{n} \sum_{i=1}^{n} |x_{(i)} - z| - \frac{1}{n^2} \sum_{i=1}^{n-1} i(n - i) \left(x_{(i+1)} - x_{(i)}\right), \tag{21}$$

where $x_{(1)} \leq x_{(2)} \leq \cdots \leq x_{(n)}$ are the sorted predictive samples. The first term measures the average absolute error between the sorted samples and the observation $z$, while the second term provides a linear-time estimate of the expected pairwise absolute differences between samples, avoiding the quadratic cost of a double sum. In this paper, we computed the empirical CRPS using Eq. (21).

For a single forecast instance, we compute $\text{CRPS}_{\text{mean}}$ as the average CRPS across all time series and prediction steps:

$$\text{CRPS}_{\text{mean}} = \mathbb{E}_{i,t} \left[\text{CRPS}\left(F_{i,t}, z_{i,t}\right)\right], \tag{22}$$

where $F_{i,t}$ denotes the predictive distribution for $z_{i,t}$, represented by its empirical CDF. Since CRPS evaluates one marginal distribution at a time, it does not capture joint dependencies across series. To address this, we also compute $\text{CRPS}_{\text{sum}}$ (Salinas et al., 2019; Drouin et al., 2022; Ashok et al., 2024), which aggregates both forecasted and observed values across all time series and applies CRPS to the resulting sums:

$$\text{CRPS}_{\text{sum}} = \mathbb{E}_t \left[\text{CRPS}\left(F_t, \sum_i z_{i,t}\right)\right], \tag{23}$$

where $F_t$ is the empirical distribution formed by summing prediction samples across all time series.

### C.5.2 Energy Score

The Energy Score (ES) generalizes the CRPS to evaluate distributional forecasts of vector-valued random variables, making it a suitable multivariate metric for this paper:

$$\widehat{\text{ES}}(F, \mathbf{z}) = \frac{1}{n} \sum_{i=1}^{n} \|\boldsymbol{x}_i - \mathbf{z}\|^{\beta} - \frac{1}{2n^2} \sum_{i=1}^{n} \sum_{j=1}^{n} \|\boldsymbol{x}_i - \boldsymbol{x}_j\|^{\beta}, \tag{24}$$

where $\|\cdot\|$ denotes the Euclidean norm, $\boldsymbol{x}_i$ and $\boldsymbol{x}_j$ are samples from the predictive distribution, and $\mathbf{z}$ is the observed vector. In this paper, we set $\beta = 1$, following Ashok et al. (2024). To aggregate over the prediction horizon, we compute the Frobenius norm of the forecast matrix $\|\mathbf{z}_{t+1:t+Q}\|_F$ in practice.

## D Additional Results

### D.1 Synthetic Data Experiment

We design a controlled noise experiment based on the example shown in §4.1 to evaluate the robustness of different proper scoring rules when estimating parameters of a Gaussian distribution in the presence of contaminated data. The experiment focuses on a two-dimensional multivariate Gaussian distribution $P = \mathcal{N}\left(\begin{bmatrix} 1 \\ -1 \end{bmatrix}, \begin{bmatrix} 1 & 0.8 \\ 0.8 & 4 \end{bmatrix}\right)$. From this distribution, we generate $N = 5000$ samples as our base dataset. To systematically study robustness properties, we introduce contamination at

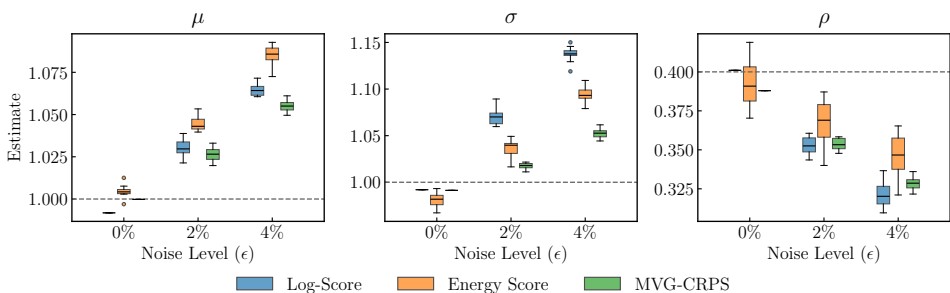

Figure A1: Parameter recovery under data contamination. Boxplots show the estimated parameters $(\mu, \sigma, \rho)$ of a bivariate Gaussian distribution using three proper scoring rules across varying contamination levels. Dashed lines indicate the ground truth values. Each boxplot summarizes estimates from 10 independent runs with different random seeds for contamination.

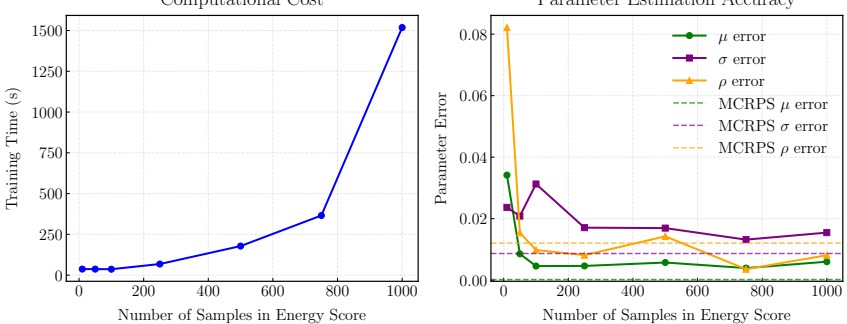

Figure A2: Computational cost versus parameter estimation accuracy for the energy score with varying sample sizes. The left panel shows training time across different numbers of Monte Carlo samples, while the right panel displays absolute errors in parameter estimates $(\mu, \sigma, \rho)$, with dashed lines indicating the corresponding MVG-CRPS reference values.

varying levels $\epsilon \in 0\%, 2\%, 4\%$ by randomly selecting $\epsilon$ proportion of individual data points and adding a fixed offset of $+3.0$ to introduce outliers.

This experiment compares three proper scoring rules for parameter estimation: the log-score; the energy score, implemented using a Monte Carlo approximation with 500 samples and $\beta = 1.0$; and the proposed MVG-CRPS. For each method and contamination level, we estimate three key parameters of the predictive distribution: $\mu$ (location), $\sigma$ (scale), and $\rho$ (correlation) in

$$Q = \mathcal{N}\left(\begin{bmatrix} \mu \\ -1 \end{bmatrix}, \begin{bmatrix} \sigma^2 & 2\rho\sigma \\ 2\rho\sigma & 4 \end{bmatrix}\right).$$

To ensure that parameter estimates remain within valid ranges, we apply a `softplus` transformation to $\sigma$ and a `tanh` transformation to $\rho$, thereby constraining them to appropriate domains.

Optimization is performed using the Adam optimizer with method-specific learning rates: $3 \times 10^{-3}$ for the log-score and MVG-CRPS, and $1 \times 10^{-2}$ for the energy score. The number of training iterations also varies: 1000 for the log-score and MVG-CRPS, and 500 for the energy score. These hyperparameters were selected based on preliminary experiments using a validation dataset and a grid search procedure to ensure a fair comparison across methods. To assess statistical significance, we conduct 10 independent runs with different random seeds for each configuration, allowing us to examine the distribution of parameter estimates across trials.

Parameter recovery accuracy is evaluated by comparing the estimated values against the ground truth. We visualize the results using boxplots, which illustrate the distribution of estimates across runs for each method and contamination level (Fig. A1). Across all three parameters, MVG-CRPS consistently yields the most accurate and stable estimates as noise increases. For the location parameter $\mu$ and the scale $\sigma$, MVG-CRPS maintains estimates closest to the true value with minimal spread, whereas both log-score and energy score drift upward under contamination. For the correlation $\rho$, noise leads to downward bias for all methods, but MVG-CRPS strikes the best balance between bias and variability. The energy score appears stable under contamination, but this stability follows from its limited sensitivity to changes in correlation, as shown in Fig. 2. Overall, MVG-CRPS shows greater robustness than the log-score and more consistent estimates than the energy score because it does not rely on Monte Carlo sampling.

Using the same example, we conducted a controlled study to examine the trade-off between computational cost and parameter estimation accuracy when using the ES with varying sample sizes. As shown in Fig. A2, training time increases monotonically with sample size due to the pairwise distance computations required by the ES. Estimation errors generally decrease with more samples but exhibit diminishing returns beyond a certain threshold (typically 100–200 samples). For reference, we include MVG-CRPS, which avoids sampling and maintains constant computational cost. Notably, even with large sample sizes (e.g., 1000), the ES does not outperform MVG-CRPS in estimation accuracy.

## D.2 Other Metrics for Multivariate Autoregressive Forecasting

The results for $\text{CRPS}_{\text{mean}}$ and ES in the multivariate autoregressive forecasting task are reported in Table A3 and Table A4, respectively. The performance of MVG-CRPS is consistent with the results reported for $\text{CRPS}_{\text{sum}}$ in Table 1.

## D.3 Univariate Seq2Seq Forecasting

The results for the univariate Seq2Seq forecasting task, presented in Table A5, Table A6, and Table A7, are consistent with those from the multivariate autoregressive task. Overall, MVG-CRPS demonstrates improved accuracy compared to both the log-score and the energy score.

Figure A3 visualizes the output covariance matrices from models trained with different loss functions. Similar to the multivariate autoregressive task, the model trained with the log-score exhibits higher variance and covariance values, indicating greater uncertainty that may reduce forecast reliability. The figure illustrates the evolution of daily covariance in the hourly `traffic` dataset, shaped by both the prediction lead time and the time of day. Uncertainty tends to increase during rush hours and at longer forecast horizons. In contrast, the model trained with MVG-CRPS captures these temporal patterns while being less sensitive to extreme values, resulting in more stable estimates.

Table A3: Comparison of $\mathrm{CRPS_{mean}}$ across different scoring rules in the multivariate autoregressive forecasting task. The best scores are in boldface. MVG-CRPS scores are underlined when they are not the best overall but exceed the log-score.

| | VAR | GPVar | | | Transformer | | |
| --- | --- | --- | --- | --- | --- | --- | --- |
| | | log-score | energy score | MVG-CRPS | log-score | energy score | MVG-CRPS |
| elec_au | N/A | 0.1261±0.0009 | **0.0887±0.0004** | 0.0967±0.0008 | 0.1633±0.0005 | 0.1492±0.0006 | **0.0793±0.0004** |
| cif_2016 | 1.0000±0.0000 | 0.1445±0.0006 | 0.1690±0.0005 | **0.1387±0.0006** | 0.1611±0.0010 | 0.1470±0.0008 | **0.1178±0.0003** |
| electricity | 0.1598±0.0007 | **0.0601±0.0004** | 0.0772±0.0003 | 0.0623±0.0002 | **0.0600±0.0002** | 0.0705±0.0003 | 0.0638±0.0002 |
| elec_weekly | 0.1237±0.0009 | 0.1427±0.0023 | **0.0676±0.0008** | 0.0878±0.0026 | 0.0964±0.0022 | 0.0726±0.0010 | **0.0697±0.0012** |
| exchange_rate | 0.0070±0.0000 | 0.0204±0.0004 | 0.0094±0.0002 | **0.0065±0.0001** | 0.0112±0.0002 | **0.0102±0.0002** | 0.0115±0.0003 |
| kdd_cup | N/A | 0.3474±0.0008 | 0.3395±0.0011 | **0.2972±0.0010** | 0.2959±0.0008 | 0.4303±0.0022 | **0.2282±0.0005** |
| m1_yearly | N/A | 0.4397±0.0041 | 0.4801±0.0022 | 0.3566±0.0029 | 0.5344±0.0109 | **0.3291±0.0047** | 0.4563±0.0111 |
| m3_yearly | N/A | 0.3607±0.0084 | 0.2186±0.0042 | **0.1423±0.0053** | 0.3156±0.0102 | 0.4050±0.0061 | **0.2325±0.0094** |
| nn5_daily | 0.2446±0.0002 | **0.1525±0.0002** | 0.1551±0.0002 | 0.1540±0.0002 | 0.1500±0.0002 | 0.1453±0.0001 | **0.1410±0.0001** |
| saugeenday | N/A | 0.4040±0.0047 | **0.3733±0.0048** | 0.3941±0.0055 | 0.3771±0.0088 | **0.3689±0.0053** | 0.3705±0.0047 |
| sunspot | N/A | 18.7115±1.3296 | 23.3988±0.9662 | **17.2438±0.5833** | 39.7454±1.4841 | **16.6556±0.6167** | 22.6495±0.6752 |
| tourism | 0.1444±0.0007 | 0.2369±0.0027 | 0.2424±0.0010 | **0.2223±0.0017** | 0.2290±0.0010 | **0.2220±0.0016** | 0.2313±0.0017 |
| traffic | 19.9208±0.0495 | **0.1357±0.0002** | 0.1367±0.0001 | 0.1415±0.0001 | 0.1185±0.0001 | 0.1327±0.0001 | **0.1174±0.0001** |
| **Avg. Rank** | | 2.23 | 2.23 | **1.54** | 2.46 | 1.92 | **1.62** |

Table A4: Comparison of ES across different scoring rules in the multivariate autoregressive forecasting task. The best scores are in boldface. MVG-CRPS scores are underlined when they are not the best overall but exceed the log-score.

| | VAR | GPVar | | | Transformer | | |
| --- | --- | --- | --- | --- | --- | --- | --- |
| | | log-score | energy score | MVG-CRPS | log-score | energy score | MVG-CRPS |
| elec_au $(\times 10^3)$ | N/A | 5.4013±0.0372 | **3.9136±0.0177** | 4.1508±0.0283 | 7.0039±0.0219 | 6.3135±0.0243 | **3.5217±0.0150** |
| cif_2016 $(\times 10^3)$ | 125.6177±0.0000 | 4.2733±0.0218 | 4.9329±0.0161 | **4.1677±0.0198** | 4.6316±0.0270 | 4.1063±0.0241 | **3.5559±0.0145** |
| elec $(\times 10^4)$ | 10.4788±0.0757 | 3.3124±0.0580 | 4.8317±0.0434 | **3.2435±0.0300** | **3.4724±0.0229** | 4.3757±0.0414 | 3.9672±0.0374 |
| elec_weekly $(\times 10^7)$ | 2.2191±0.0308 | 2.5724±0.0799 | **0.8948±0.02344** | 1.4040±0.0887 | 1.5463±0.0582 | **0.9338±0.0308** | 0.9985±0.0360 |
| exchange_rate | 0.1301±0.0002 | 0.3972±0.0074 | 0.1895±0.0034 | **0.1216±0.0013** | 0.2136±0.0026 | **0.1774±0.0026** | 0.2040±0.0045 |
| kdd_cup $(\times 10^2)$ | N/A | 4.7575±0.0186 | 4.3981±0.0164 | **4.0719±0.0180** | 4.2809±0.0134 | 5.9466±0.0427 | **3.1788±0.0122** |
| m1_yearly $(\times 10^4)$ | N/A | 7.3860±0.0789 | 7.7576±0.0335 | **6.1985±0.0505** | 8.7079±0.1760 | **5.7774±0.0755** | 7.5130±0.1784 |
| m3_yearly $(\times 10^3)$ | N/A | 3.6113±0.0703 | 2.2147±0.0427 | **1.4775±0.0495** | 3.1996±0.0995 | 4.0982±0.0621 | **2.4253±0.0914** |
| nn5_daily $(\times 10^2)$ | 4.9419±0.0056 | **3.3001±0.0050** | 3.3004±0.0052 | 3.3934±0.0045 | 3.2546±0.0033 | 3.1622±0.0045 | **3.0996±0.0025** |
| saugeenday $(\times 10^2)$ | N/A | 1.8098±0.0231 | **1.7135±0.0150** | 1.9400±0.0208 | **1.5780±0.0183** | 1.5883±0.0108 | 1.8043±0.0204 |
| sunspot $(\times 10)$ | N/A | 2.7737±0.1195 | 3.1658±0.0792 | **2.6195±0.1003** | 5.4893±0.1132 | **2.3153±0.0467** | 3.2663±0.0745 |
| tourism $(\times 10^5)$ | 3.5958±0.0354 | 6.1085±0.1132 | 5.6774±0.0493 | **5.2111±0.0896** | 5.0645±0.0526 | **4.7502±0.0585** | 5.2702±0.0853 |
| traffic_nips | 3358.5004±10.7535 | 2.2924±0.0034 | **2.1140±0.0023** | 2.2916±0.0015 | 2.2043±0.0012 | 2.2250±0.0018 | **2.2000±0.0018** |
| **Avg. Rank** | | 2.46 | 2.00 | **1.54** | 2.38 | 1.92 | **1.69** |

Figure A4 further compares probabilistic forecasts on the `m4_hourly` dataset. The model trained with MVG-CRPS produces narrower and better-calibrated prediction intervals than the log-score-trained model, particularly for time series with clear cyclical patterns. It also achieves higher accuracy at longer forecast horizons. These results indicate that MVG-CRPS enhances both robustness and calibration, leading to more accurate and reliable forecasts.

Table A5: Comparison of $\text{CRPS}_{\text{sum}}$ across different scoring rules in the univariate Seq2Seq forecasting task. The best scores are in boldface. MVG-CRPS scores are underlined when they are not the best overall but exceed the log-score.

|  | N-HiTS | | |
| --- | --- | --- | --- |
|  | log-score | energy score | MVG-CRPS |
| covid | 0.1297±0.0048 | N/A | **0.1011±0.0022** |
| elec_hourly | 0.0470±0.0008 | N/A | **0.0398±0.0004** |
| electricity | 0.0409±0.0003 | 0.0378±0.0006 | **0.0372±0.0003** |
| exchange_rate | 0.0089±0.0005 | 0.0060±0.0002 | **0.0053±0.0002** |
| m4_hourly | 0.0649±0.0007 | 0.0595±0.0005 | **0.0399±0.0007** |
| nn5_daily | 0.0571±0.0003 | 0.0876±0.0006 | **0.0569±0.0004** |
| pedestrian | 0.7985±0.0511 | 0.9110±0.0210 | **0.5296±0.0071** |
| saugeenday | 0.4804±0.0150 | 0.4372±0.0100 | **0.3864±0.0035** |
| taxi_30min | 0.0496±0.0002 | 0.0603±0.0002 | **0.0449±0.0001** |
| traffic | 0.2065±0.0007 | **0.0815±0.0001** | 0.0832±0.0002 |
| uber_hourly | 0.7027±0.0209 | 0.6461±0.0052 | **0.5380±0.0033** |
| wiki | 0.0660±0.0011 | **0.0429±0.0003** | 0.0465±0.0004 |
| **Avg. Rank** | 2.70 | 2.10 | **1.20** |

Table A6: Comparison of $\text{CRPS}_{\text{mean}}$ across different scoring rules in the univariate Seq2Seq forecasting task. The best scores are in boldface. MVG-CRPS scores are underlined when they are not the best overall but exceed the log-score.

|  | N-HiTS | | |
| --- | --- | --- | --- |
|  | log-score | energy score | MVG-CRPS |
| covid | 0.2076±0.0018 | 0.1440±0.0013 | **0.1022±0.0012** |
| elec_hourly | 0.0903±0.0005 | 0.1189±0.0004 | **0.0874±0.0003** |
| electricity | 0.0671±0.0002 | 0.0913±0.0002 | **0.0635±0.0001** |
| exchange_rate | 0.0173±0.0004 | 0.0077±0.0001 | **0.0073±0.0001** |
| m4_hourly | 0.1599±0.0003 | 0.1762±0.0007 | **0.1093±0.0005** |
| nn5_daily | 0.1964±0.0006 | **0.1588±0.0002** | 0.1846±0.0008 |
| pedestrian | 1.0856±0.0262 | 0.9254±0.0105 | **0.7328±0.0076** |
| saugeenday | 0.4804±0.0150 | 0.4372±0.0100 | **0.3864±0.0035** |
| taxi_30min | 0.3853±0.0001 | 0.3939±0.0001 | **0.3219±0.0000** |
| traffic | 0.2514±0.0004 | 0.1726±0.0001 | **0.1583±0.0001** |
| uber_hourly | 0.9630±0.0272 | 0.8229±0.0062 | **0.6852±0.0040** |
| wiki | 0.4160±0.0006 | 0.2824±0.0003 | **0.2656±0.0002** |
| **Avg. Rank** | 2.67 | 2.25 | **1.08** |

## D.4 HYPERPARAMETER SENSITIVITY

To ensure a fair comparison, our main experiments used fixed hyperparameters across all loss functions. However, since certain hyperparameters such as learning rate and rank do not affect the model architecture, we performed grid searches over learning rates $10^{-2}, 10^{-3}, 10^{-4}$ and rank values $10, 20, 30$ for each dataset. The optimal configuration was selected based on validation performance for each combination of loss function, model group, and dataset. The results are presented in Table A8 and Table A9. With tuned hyperparameters, the MVG-CRPS still achieves the best average rank.

## D.5 CONTROLLED OUTLIER EXPERIMENT

We conducted an additional experiment by injecting synthetic outliers into the training data. Specifically, a fixed proportion of observations for each sensor was perturbed with large noise ($\pm 5\times$ the sensor's standard deviation). The test data remained clean to isolate the impact of training-time

Table A7: Comparison of ES across different scoring rules in the univariate Seq2Seq forecasting task. The best scores are in boldface. MVG-CRPS scores are underlined when they are not the best overall but exceed the log-score.

| | N-HiTS | | |
| --- | --- | --- | --- |
| | log-score | energy score | MVG-CRPS |
| covid $(\times 10^5)$ | 2.1220±0.0304 | N/A | **0.9401±0.0186** |
| elec_hourly $(\times 10^5)$ | 0.9283±0.0161 | N/A | **0.9088±0.0079** |
| elec $(\times 10^5)$ | 0.2535±0.0018 | 0.3123±0.0019 | **0.2431±0.0020** |
| exchange_rate | 0.2876±0.0055 | 0.1272±0.0022 | **0.1240±0.0022** |
| m4_hourly $(\times 10^4)$ | 0.2852±0.0026 | 0.2890±0.0029 | **0.2423±0.0027** |
| nn5_daily $(\times 10^3)$ | 0.4170±0.0018 | **0.3272±0.0005** | 0.3958±0.0021 |
| pedestrian $(\times 10^3)$ | 1.1571±0.0177 | 0.9746±0.0081 | **0.8337±0.0066** |
| saugeenday $(\times 10^2)$ | **1.6690±0.0391** | 1.7752±0.0216 | 1.7698±0.0129 |
| taxi_30min $(\times 10^2)$ | 6.9676±0.0045 | 6.7906±0.0058 | **5.6679±0.0004** |
| traffic | 3.6810±0.0136 | 2.2524±0.0018 | **2.2200±0.0022** |
| uber_hourly | 6.3252±0.1785 | 5.4214±0.0326 | **4.2826±0.0320** |
| wiki $(\times 10^6)$ | 1.1535±0.0047 | 0.9352±0.0069 | **0.9338±0.0083** |
| **Avg. Rank** | 2.60 | 2.20 | **1.20** |

Table A8: Comparison of $\mathrm{CRPS_{mean}}$ across different scoring rules in the multivariate autoregressive forecasting task. The best scores are in boldface. MVG-CRPS scores are underlined when they are not the best overall but exceed the log-score. The results are obtained using models with the best hyperparameters (learning rate and rank), selected for each loss function, model group, and dataset based on validation performance. For the energy score, hyperparameter tuning was omitted due to extended training time.

| | VAR | GPVar | | | Transformer | | |
| --- | --- | --- | --- | --- | --- | --- | --- |
| | | log-score | energy score | MVG-CRPS | log-score | energy score | MVG-CRPS |
| elec_au | N/A | 0.0437±0.0004 | 0.0887±0.0004 | **0.0280±0.0002** | 0.1158±0.0005 | 0.1492±0.0006 | 0.1410±0.0004 |
| cif_2016 | 1.0000±0.0000 | 0.1444±0.0004 | 0.1690±0.0005 | **0.1275±0.0003** | 0.1217±0.0005 | 0.1470±0.0008 | **0.1201±0.0002** |
| electricity | 0.1598±0.0007 | **0.0601±0.0004** | 0.0772±0.0003 | 0.0665±0.0004 | **0.0605±0.0003** | 0.0705±0.0003 | 0.0650±0.0002 |
| elec_weekly | 0.1237±0.0009 | 0.1128±0.0014 | **0.0676±0.0008** | 0.1046±0.0025 | 0.1000±0.0020 | **0.0726±0.0010** | 0.1061±0.0013 |
| exchange_rate | 0.0070±0.0000 | **0.0071±0.0001** | 0.0094±0.0002 | 0.0093±0.0002 | 0.0131±0.0003 | 0.0102±0.0002 | 0.0161±0.0002 |
| kdd_cup | N/A | 0.3274±0.0015 | 0.3395±0.0011 | **0.2861±0.0004** | 0.2865±0.0012 | 0.4303±0.0022 | **0.2291±0.0010** |
| m1_yearly | N/A | 0.4883±0.0088 | 0.4801±0.0022 | **0.3333±0.0015** | 0.5394±0.0111 | **0.3291±0.0047** | 0.4420±0.0070 |
| m3_yearly | N/A | 0.3606±0.0133 | 0.2186±0.0042 | **0.1423±0.0053** | 0.3658±0.0097 | 0.4050±0.0061 | **0.2964±0.0136** |
| nn5_daily | 0.2446±0.0002 | **0.1474±0.0002** | 0.1551±0.0002 | 0.1510±0.0001 | 0.1466±0.0001 | 0.1453±0.0001 | **0.1430±0.0001** |
| saugeenday | N/A | 0.3715±0.0032 | 0.3733±0.0048 | **0.3600±0.0053** | 0.3756±0.0055 | **0.3689±0.0053** | 0.3831±0.0032 |
| sunspot | N/A | **10.7124±0.4618** | 23.3988±0.9662 | 16.1930±0.5734 | 14.4194±0.5650 | 16.6556±0.6167 | **13.1737±0.6602** |
| tourism | 0.1444±0.0007 | 0.2492±0.0015 | 0.2424±0.0010 | **0.1193±0.0020** | 0.2258±0.0020 | 0.2220±0.0016 | **0.2082±0.0014** |
| traffic | 19.9208±0.0495 | 0.1534±0.0002 | **0.1367±0.0001** | 0.1415±0.0001 | 0.1422±0.0001 | 0.1327±0.0001 | **0.1152±0.0000** |
| **Avg. Rank** | | 2.08 | 2.46 | **1.46** | 2.15 | 2.08 | **1.77** |

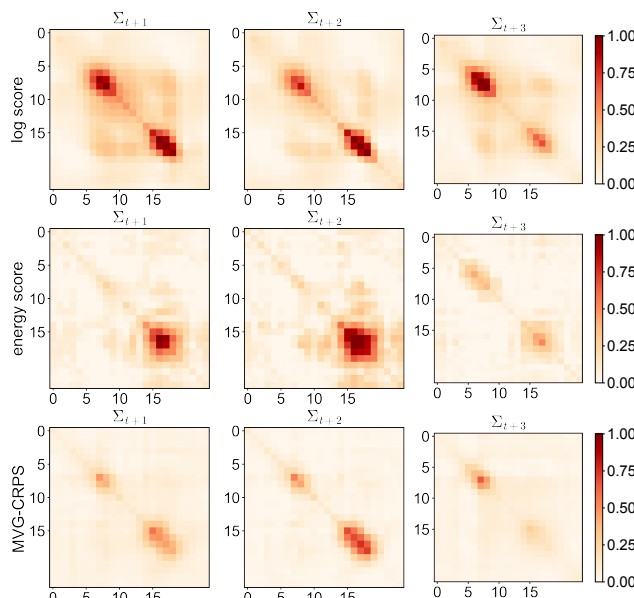

Figure A3: Comparison of output covariance matrices $\Sigma_i$ from N-HiTS on the `traffic` dataset. For visual clarity, covariance values are clipped between 0 and 1.0.

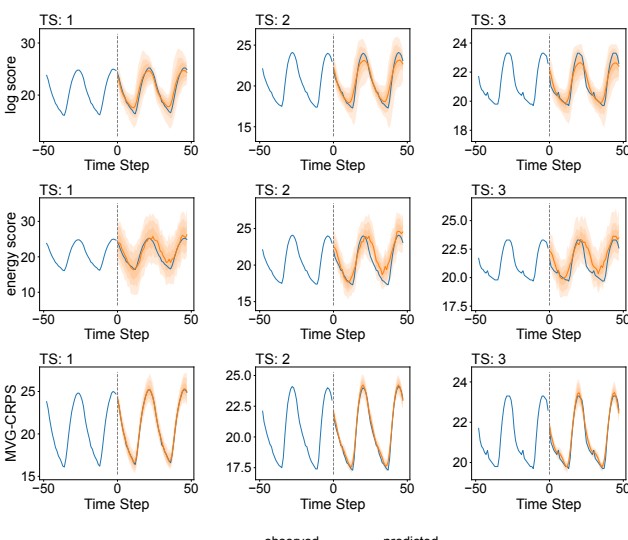

Figure A4: Comparison of probabilistic forecasts from N-HiTS on the `m4_hourly` dataset.

contamination. Results in Fig. A5 indicate that models trained with the log-score degrade rapidly under such noise, whereas the MVG-CRPS demonstrates greater robustness.

Table A9: Comparison of ES across different scoring rules in the multivariate autoregressive forecasting task. The best scores are in boldface. MVG-CRPS scores are underlined when they are not the best overall but exceed the log-score. The results are obtained using models with the best hyperparameters (learning rate and rank), selected for each loss function, model group, and dataset based on validation performance. For the energy score, hyperparameter tuning was omitted due to extended training time.

| | VAR | GPVar | | | Transformer | | |
|---|---|---|---|---|---|---|---|
| | | log-score | energy score | MVG-CRPS | log-score | energy score | MVG-CRPS |
| elec_au ($\times 10^3$) | N/A | 1.9601±0.0200 | 3.9136±0.0177 | **1.2546±0.0066** | **4.9064±0.0215** | 6.3135±0.0243 | 5.9514±0.0150 |
| cif_2016 ($\times 10^3$) | 125.6177±0.0000 | 4.3478±0.0127 | 4.9329±0.0161 | **3.8815±0.0072** | 3.6976±0.0203 | 4.6316±0.0270 | **3.5888±0.0118** |
| elec ($\times 10^4$) | 10.4788±0.0757 | **3.6854±0.0973** | 4.8317±0.0434 | 3.6913±0.0353 | 4.9963±0.0371 | **3.4724±0.0229** | 4.6774±0.0544 |
| elec_weekly ($\times 10^7$) | 2.2191±0.0308 | 1.9808±0.0774 | **0.8948±0.0234** | 1.2270±0.0539 | 1.5074±0.0600 | 1.5463±0.0582 | **1.0231±0.0402** |
| exchange_rate | 0.1301±0.0002 | 0.2166±0.0061 | 0.1895±0.0034 | **0.1519±0.0018** | 0.2317±0.0042 | 0.2136±0.0026 | **0.1569±0.0032** |
| kdd_cup ($\times 10^2$) | N/A | 4.7575±0.0186 | **4.3981±0.0164** | 5.0382±0.0142 | 5.2922±0.0202 | 4.2809±0.0134 | **3.2651±0.0134** |
| m1_yearly ($\times 10^4$) | N/A | 8.1941±0.1388 | 7.7576±0.0335 | **5.9567±0.0210** | 8.7995±0.1777 | 8.7079±0.1777 | **7.2322±0.0979** |
| m3_yearly ($\times 10^3$) | N/A | 3.6966±0.1408 | 2.2147±0.0427 | **1.4775±0.0495** | 3.7233±0.0925 | 3.1996±0.0995 | **2.9483±0.1275** |
| nn5_daily ($\times 10^2$) | 4.9419±0.0056 | **3.1966±0.0044** | 3.3004±0.0052 | 3.3303±0.0031 | **3.1311±0.0038** | 3.2546±0.0033 | 3.1725±0.0033 |
| saugeenday ($\times 10^2$) | N/A | **1.6529±0.0150** | 1.7135±0.0150 | 1.7678±0.0188 | 1.6434±0.0160 | **1.5780±0.0183** | 1.6426±0.0220 |
| sunspot ($\times 10$) | N/A | **1.7726±0.0430** | 3.1658±0.0792 | 2.5742±0.0651 | 2.1717±0.0468 | 5.4893±0.1132 | **1.9724±0.0363** |
| tourism ($\times 10^5$) | 3.5958±0.0354 | 6.5310±0.0670 | 5.6774±0.0493 | **2.8103±0.1048** | 6.0582±0.1162 | 5.0645±0.0526 | **4.5365±0.0662** |
| traffic_nips | 3358.5004±10.7535 | 2.4690±0.0026 | **2.1140±0.0023** | 2.2967±0.0012 | 2.2314±0.0016 | 2.2043±0.0012 | **2.1626±0.0020** |
| **Avg. Rank** | | 2.15 | 2.08 | **1.77** | 2.46 | 2.23 | **1.31** |

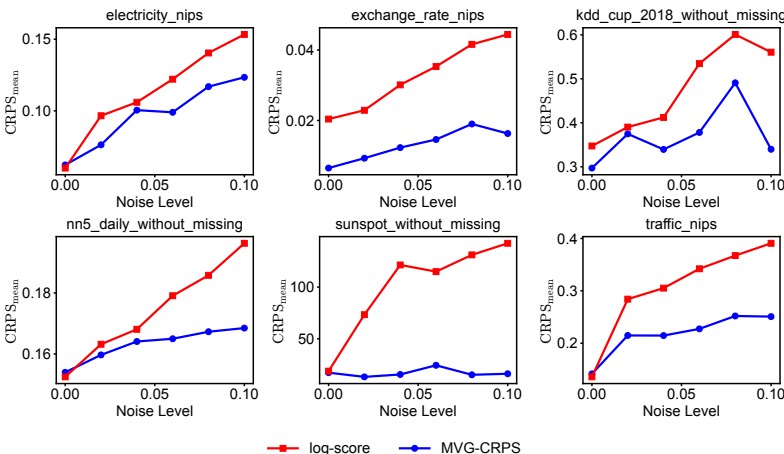

Figure A5: Controlled outlier experiment using GPVar. A fixed proportion of training samples per sensor is perturbed by adding large noise.

