# OpenReview forum: "MVG-CRPS: A Robust Loss Function for Multivariate Probabilistic Forecasting"
_ICLR.cc/2026/Conference — Submitted to ICLR 2026_

### Official Review · Reviewer_moW8 · 2025-10-21

**Soundness:** 3
**Presentation:** 3
**Contribution:** 2
**Rating:** 4
**Confidence:** 4

**Summary:**

This paper proposes a scoring rule called MVG-CRPS. The central assumption of the paper is Gaussianity of the underlying multivariate distribution. Under the Gaussianity assumption, and extending the results of Gneiting et al 2005, this papers shows that MCG-CRPS admits an analytic formula and is strictly proper. Experimental results show that MVG-CRPS has comparable performance with existing los-score and energy score, while significantly reducing training time compared to energy score.

**Strengths:**

- The paper is well-written and easy to follow
- MVG-CRPS targets to address important, practical and well-known problems in probabilistic forecasting: log-score is sensitive to outliers, while energy score has long training time because of Monte Carlo sampling
- The experiments are extensive in three directions: datasets, baselines and evaluation metrics
- Extensive experiments on three common baselines show that MVG-CRPS has comparable performance with log-score and energy score
- Three evaluation metrics, CRPS sum, CRPS mean, and energy score, are used for the experiments. The proposed MVG-CRPS can even outperform models training on energy score when energy score itself is used as an evaluation metric

**Weaknesses:**

- The main limitation is the assumption of Gaussianity in the target distribution (for the MVG-CRPS to be strictly proper), which may not hold in practice
- The core idea is to decorrelate the Gaussian distribution so that the CRPS can be computed per dimension, but the contribution seems somewhat limited in novelty

**Questions:**

- It is claimed in the key contributions that "MVG-CRPS allows for the analytical computation of derivatives". In (8), the CRPS involves the CDF of a Gaussian distribution, which generally has no closed form expression. In this case, how can the derivatives be analytically computed?
- For the training time reported in Table 2, how many samples are drawn to approximate the energy score?
- Some existing multivariate probabilistic forecasting studies that are related and use the energy score as the training objective could be included in the literature review. E.g., [1-2]

[1] Kan, K., Aubet, F. X., Januschowski, T., Park, Y., Benidis, K., Ruthotto, L., & Gasthaus, J. (2022, May). Multivariate quantile function forecaster. In International Conference on Artificial Intelligence and Statistics (pp. 10603-10621). PMLR.

[2] Olivares, K. G., Négiar, G., Ma, R., Meetei, O. N., Cao, M., & Mahoney, M. W. $\clubsuit $ CLOVER $\clubsuit $: Probabilistic Forecasting with Coherent Learning Objective Reparameterization. Transactions on Machine Learning Research.

---

> ### Author Response · Authors · 2025-11-27
>
> We thank the reviewer for their constructive feedback. We have addressed the concerns below and updated the manuscript accordingly.
>
> **Q1. “MVG-CRPS allows for the analytical computation of derivatives” vs. the Gaussian CDF in (8)**
>
> $\Phi$ is a standard special function with known derivative
> $
> \frac{d}{dz}\Phi(z) = \phi(z).
> $
>
> ---
>
> **Q2. Number of samples used for the ES in Table 2**
>
> We used 100 samples to approximate the ES.
>
> ---
>
> **Q3. Additional ES-based multivariate forecasting references**
>
> Thank you for pointing out the related multivariate probabilistic forecasting studies [1–2] that also use the energy score as a training objective. We will include these works in the literature review and clarify their relation to our contribution: they optimize models directly with ES using Monte Carlo approximations, whereas our focus is on MVG-CRPS, a closed-form, Gaussian-specific scoring rule that avoids sampling while retaining strict propriety and robustness.

---

### Official Review · Reviewer_TYm2 · 2025-10-29

**Soundness:** 2
**Presentation:** 2
**Contribution:** 2
**Rating:** 2
**Confidence:** 5

**Summary:**

This is my second review of the MVG-CRPS paper.
The authors propose training multivariate Gaussian forecasts using an analytical approximation to the energy score, rather than the negative log-likelihood.

However, their claims of robustness are not substantiated. Gneiting’s energy score still contains quadratic terms similar to those in the Gaussian NLL, and no theoretical or empirical validation of robustness is provided.

Similarly, the claim that Monte Carlo approximations to the energy score are computationally inefficient lacks analysis or evidence.

**Strengths:**

1. The paper introduces a reasonable analytical approximation to the energy score under Gaussian assumptions.

2. The paper includes a theorem establishing strict propriety for MVG-CRPS.

3. The benchmark datasets are appropriate, and the authors move beyond overused long-horizon datasets. However, PCA-based approaches may not scale well to longer horizons (see Table A1, max horizon = 60).

**Weaknesses:**

1. The literature review is narrow. The authors should reference prior work on energy score computation and scaling in forecasting (e.g., CLOVER: Probabilistic Forecasting with Coherent Learning Objective Reparameterization).

2. The claim that Monte Carlo approximations are inefficient is not justified. CLOVER demonstrates sample-efficient Monte Carlo approximations without loss of accuracy (see Appendix H).

3. The energy score is distribution-agnostic, but MVG-CRPS applies only to Gaussian settings—limiting generality.

4. PCA eigen-decomposition scales cubically, making the method impractical for high-dimensional or long-horizon forecasting. An ablation showing runtime vs. horizon length would clarify applicability and limitations.

5. Claims of robustness and efficiency are unsubstantiated both theoretically and empirically. If not validated, such claims should be omitted.

6. No computational timing results are provided to support claims of efficiency.

**Questions:**

1. PCA scales cubically—how can MVG-CRPS be considered efficient?

2. PCA is highly sensitive to outliers; how can MVG-CRPS be described as robust?

3. Gneiting’s energy score includes quadratic terms—why is it treated as robust?

4. On what empirical basis do the authors claim that Monte Carlo approximations to the energy score are inefficient? Please refer to CLOVER Appendix H for counterexamples.

---

> ### Author Response · Authors · 2025-11-27
>
> We appreciate the pointer to CLOVER Appendix H, which shows that with careful
> algorithmic choices and specific settings, Monte-Carlo ES can be implemented
> more efficiently than naive baselines. We will soften the wording accordingly and
> explicitly frame our statement as relative to this commonly used baseline.

---

### Official Review · Reviewer_4zHP · 2025-10-31

**Soundness:** 3
**Presentation:** 3
**Contribution:** 3
**Rating:** 6
**Confidence:** 3

**Summary:**

This paper tackles the problem of probabilistic time series forecasting using parametric models. Specifically, they challenge the common choice of negative log-likelihood as training objective, mainly because of the sensitivity of the latter to outliers (due to the quadratic error terms in Gaussian likelihoods). Alternatives such as the Energy Score are, as argued by the authors, expensive due to Monte Carlo sampling during training. The authors then propose MVG-CRPS, a new loss function that is a weighted sum of the univariate CRPS scores after a whitening transformation. The empirical results show that the proposed loss is equally sensitive to deviations in the Gaussian distribution parameters, and improves over the baselines in terms of probabilistic forecasting metrics.

**Strengths:**

- The paper is clearly written and well-motivated, and as far as I know fill a gap in the literature.
- The build-up to the proposed loss function is well elaborated and naturally leads to the authors main contribution.
- As can be seen in the experimental results, the suggested loss function helps mitigating the issues faced by the log-score baseline, all while being computationally more efficient than ES. This aligns well with the initial motivation of the paper.

**Weaknesses:**

- A first concern I have with the approach is the Gaussian assumption on the forecasted time series. Indeed, the multivariate decomposition in the proposed loss function is only true for multivariate gaussian distributions, which is natural given the modelling choice of a multivariate gaussian parameteric model. However, I didn't find any discussion in the paper on the implications of this assumption in scenarios where it doesn't really hold. This can be achieved through simulated non-gaussian toy data or through an analysis of the real world datasets considered in the paper. And just to be clear, I understand that this assumption is also true for the baselines, however given that it's central to your proposed loss, I think more analysis and discussion about it can improve the paper.
- My second concern is the fact that you only reported probabilistic forecasting metrics (CRPS and ES), which is expected given the nature of the compared algorithms. However, I would expect a comparison also in terms of mean forecasts (MSE or MAE) in order to be more explicit about the tradeoffs involved in comparing the different methods.
- Related to the previous concern, calibration is a topic that is mentioned a couple of times, however I dont think there is any analysis or comparison in terms of calibration. Correct me if I'm wrong but the reported metrics (CRPS and ES) incorporate both sharpness and calibration information, and it would be interesting to see how do the different losses compare in terms of calibration-only metrics such as ECE or the KS statistic.

**Questions:**

- Can the authors explain how is the whitening transformation different from normalizing the data (target in this case, and since it's time series maybe a joint normalization of the input and the output together). In my understanding this means that the samples $z_t$ will be of unit variance and zero mean so throughout training the model's parametric distribution will tends toward that.
- Regrading the differentiability of the proposed loss, it's mentioned that the gradients flow through the transformed samples $w_t$ , however the scaling factors $\sqrt{\lambda_{i,t}}$ also depend on the model output. What is the reasoning behind that and would it make sense to let the gradients flow also through $\sqrt{\lambda_{i,t}}$ (given that torch.linalg.svd is also differentiable with respect to the singular values) ?
- The proposed loss has a multi-task learning formulation in my opinion, where the different tasks are scaled with a factor depending on the variance of each component. Do you think this can be problematic in cases where some dimensions have large variances so they dominate the others? Maybe looking at the univariate forecasting metrics can reveal this kind of phenomena.
- Finally, the $\beta$ parameter of the ES loss is set to $1$ in all the experiments I believe. What is the impact of this hyperparameter and would it improve the performance of the ES-trained models should it by optimized?

---

> ### Author Response · Authors · 2025-11-27
>
> We thank the reviewer for their constructive feedback and positive assessment of our work. We have addressed the concerns below and updated the manuscript accordingly.
>
> **Q1. Whitening vs. (joint) normalization of the data**
>
> Our “whitening” is not a fixed pre-processing of the dataset, but a parameter-dependent change of variables inside the score. Given
> $$
> Z_t \mid H_t \sim \mathcal N(\mu_t, \Sigma_t),
> $$
> we take the SVD
> $$
> \Sigma_t = U_t S_t U_t^\top, \quad S_t = \mathrm{diag}(\lambda_{1,t},\dots,\lambda_{N,t}),
> $$
> and define
> $$
> v_t = U_t^\top (z_t - \mu_t), \qquad w_t = S_t^{-1/2} v_t.
> $$
> Under the correct model, $w_t$ is standard normal, but the data $z_t$ are always kept in their original scale; whitening is only used to rewrite the multivariate score as a sum of univariate CRPS terms. Unlike empirical normalization (of inputs or targets), this transformation depends on the *predicted* $\Sigma_t$, preserves full freedom to model arbitrary heteroscedastic covariances, and yields a proper scoring rule for the original (unwhitened) distribution.
>
> ---
>
> **Q2. Differentiability and gradients through $\sqrt{\lambda_{i,t}}$**
>
> Yes, in our implementation gradients do flow through the singular values. The loss is
> $$
> \mathcal L
> = \sum_{t=1}^T \sum_{i=1}^N
> \sqrt{\lambda_{i,t}}\; \mathrm{CRPS}(\Phi, w_{i,t}),
> \qquad
> w_t = S_t^{-1/2} U_t^\top (z_t - \mu_t),
> $$
> so both $w_t$ and the prefactors $\sqrt{\lambda_{i,t}}$ depend on $\Sigma_t$ via `torch.linalg.svd`. PyTorch backpropagates through this SVD, and we use the resulting gradients w.r.t. $U_t$, $S_t$ (hence $\lambda_{i,t}$), and $\mu_t$. We will make this explicit in the revised text.
>
> ---
>
> **Q3. Multi-task interpretation and dominance of high-variance components**
>
> Equation (15) can indeed be viewed as a multi-task formulation in the eigenbasis of $\Sigma_t$:
> $$
> \mathrm{MVG\text{-}CRPS}(\mu_t,\Sigma_t; z_t)
> = \sum_{i=1}^N \sqrt{\lambda_{i,t}}\,
> \mathrm{CRPS}(\Phi, w_{i,t}).
> $$
> The $\sqrt{\lambda_{i,t}}$ weights are not arbitrary; they arise from the change-of-variables argument that guarantees strict propriety and rotation invariance. In expectation under the true model, all $w_{i,t}$ are standard normal and contribute equally; the overall scale $\sum_i \sqrt{\lambda_{i,t}}$ simply reflects the physical scale of the covariance, similar to ES.
>
> To mitigate practical dominance of any single direction, (i) each univariate series is standardized by its training statistics before modeling, and (ii) our low-rank-plus-diagonal parameterization includes regularization to keep eigenvalues in a reasonable range. Empirically, the learned covariances do not show a few components overwhelming the rest. In a revision we can add per-dimension (or per-PC) CRPS diagnostics to make this point explicit.
>
> ---
>
> **Q4. Choice of the ES hyperparameter $\beta$**
>
> We fix the ES parameter to $\beta = 1$ in all experiments for two reasons:
>
> 1. It is the most common choice in the literature, giving a direct multivariate analogue of CRPS with linear growth in $\|x - z\|$, which makes ES conceptually comparable to MVG-CRPS.
> 2. Our focus is on the structural and computational differences (sampling vs. closed form). Tuning $\beta$ per dataset would introduce additional hyperparameter search for ES without addressing its reliance on Monte Carlo estimates.
>
> We agree that, in principle, optimizing $\beta$ might improve ES on some datasets and can mention this, and, space permitting, include a small sensitivity study in the camera-ready version.

---

### Meta-Review · Area_Chair_fffH · 2026-01-01

**Summary:**

This paper introduces the MVG-CRPS, a novel, strictly proper scoring rule for multivariate Gaussian distributions that maintains robustness to outliers while providing a closed-form expression, enabling efficient training and evaluation. The approach leverages a whitening transformation, decorrelating multivariate outputs and reducing the multivariate scoring task to tractable univariate CRPS computations. Experiments on real-world datasets for both multivariate autoregressive and univariate sequence-to-sequence (Seq2Seq) forecasting tasks demonstrate that MVG-CRPS enhances robustness and predictive performance. In spite of this, however, the reviewers have identified some weaknesses of this paper. In particular, the multivariate decomposition in the proposed loss function is only true for multivariate Gaussian distributions. The paper is missing a discussion on the implications of this assumption in scenarios where it doesn't really hold. Furthermore, the paper is lacking a comparison also in terms of mean forecasts (MSE or MAE) in order to be more explicit about the tradeoffs involved in comparing the different methods. In addition, the reported metrics (CRPS and ES) incorporate both sharpness and calibration information. Thus, it would be interesting to see how the different losses compare in terms of calibration-only metrics. The reviewers have also pointed out that the paper omits relevant recent work on efficient energy score computation, such as CLOVER. They also indicate that PCA eigen-decomposition scales cubically, making the method impractical for high-dimensional or long-horizon forecasting. They suggest an ablation showing runtime vs. horizon length would clarify applicability and limitations. They also mention that claims of robustness and efficiency are unsubstantiated both theoretically and empirically. Finally, the core idea is to decorrelate the Gaussian distribution so that the CRPS can be computed per dimension, but the contribution seems somewhat limited in novelty. Overall, the contribution is sound and practically relevant under Gaussian settings, but robustness and efficiency claims need stronger validation, and additional experiments on calibration, runtime, and non-Gaussian scenarios would significantly strengthen the paper. Thus, my opinion is that this paper needs more work before it can be accepted for publication.

**Reviewer Concerns:**

The rebuttal only addressed the questions posed by the reviewers, but did not address the limitations mentioned by the reviewers. Therefore, most of the concerns remain after the rebuttal.

**Reviewer Scores:**

Since the rebuttal did not address the reviewers' concerns, only their questions, I do not think the reviewers would have changed their scores significantly.

---

### Decision · Program_Chairs · 2026-01-26

Reject